**EMBO**
*reports*

# Novel integrated multiomics analysis reveals a key role for integrin beta-like 1 in wound scarring

Sang-Eun Kim [1], Ryota Noda[1], Yu-Chen Liu [2], Yukari Nakajima[3], Shoichiro Kameoka[4], Daisuke Motooka[2,4], Seiya Mizuno [5], Satoru Takahashi [5], Kento Takaya[3], Takehiko Murase[6,7], Kazuya Ikematsu[6], Katsiaryna Tratsiakova[8], Takahiro Motoyama [8], Masahiro Nakashima [8], Kazuo Kishi [3], Paul Martin [9], Shigeto Seno [10,13]✉, Daisuke Okuzaki [2,4,13]✉ & Ryoichi Mori [1,11,12]✉

## Abstract

Exacerbation of scarring can originate from a minority fibroblast population that has undergone inflammatory-mediated genetic changes within the wound microenvironment. The fundamental relationship between molecular and spatial organization of the repair process at the single-cell level remains unclear. We have developed a novel, high-resolution spatial multiomics method that integrates spatial transcriptomics with scRNA-Seq; we identified new characteristic features of cell–cell communication and signaling during the repair process. Data from *PU.1⁻/⁻* mice, which lack an inflammatory response, combined with scRNA-Seq and Visium transcriptomics, led to the identification of nine genes potentially involved in inflammation-related scarring, including integrin beta-like 1 (*Itgbl1*). Transgenic mouse experiments confirmed that *Itgbl1*-expressing fibroblasts are required for granulation tissue formation and drive fibrogenesis during skin repair. Additionally, we detected a minority population of *Acta2^high*-expressing myofibroblasts with apparent involvement in scarring, in conjunction with *Itgbl1* expression. IL1β signaling inhibited *Itgbl1* expression in TGFβ1-treated primary fibroblasts from humans and mice. Our novel methodology reveal molecular mechanisms underlying fibroblast–inflammatory cell interactions that initiate wound scarring.

**Keywords** Inflammation; Itgbl1; Multiomics; Scarring; Skin Wound Healing
**Subject Categories** Cell Adhesion, Polarity & Cytoskeleton; Methods & Resources; Skin

## Introduction

The skin wound-healing process comprises three phases: an inflammatory phase, which is primarily characterized by inflammatory cell infiltration; a proliferation phase, which is characterized by re-epithelialization and formation of granulation tissue; and finally, a maturation phase, during which excess extracellular matrix (ECM) is degraded, and a scar is formed. These processes are controlled by cell–cell interactions and cytokine networks (Peña and Martin, 2024). While scars typically remain after wound healing, skin wounds in mouse embryos (before embryonic day 15) and late-gestation human fetuses (before the end of the second trimester) can fully regenerate without any overt sign of scarring (Hopkinson-Woolley et al, 1994; Lorenz et al, 1992). Inflammatory cells do not accumulate at the wound site in these situations, suggesting that inflammatory cells play a critical role in scar formation, but currently, the molecular mechanisms that underlie inflammation-driven scarring remain unclear.

To address this phenomenon, we previously conducted comprehensive analyses of inflammation-related mRNAs and microRNAs (miRNAs) by comparing *PU.1*-deficient (*PU.1⁻/⁻*) mice, which lack neutrophils, macrophages, mast cells, and T cells, with their wild-type (WT) counterparts (Cooper et al, 2005; de Kerckhove et al, 2018). Neonatal *PU.1⁻/⁻* mice exhibit rapid, scarless skin wound healing compared with WT mice (Martin et al, 2003). One gene revealed by this analysis was the integrin-binding protein *Spp1*, expressed by wound-infiltrating fibroblasts, which we identified as a driver of scarring; indeed, local knockdown of *Spp1* in wounds inhibited scar formation (Mori et al, 2008). Notably, only a minority fibroblast population was activated by inflammatory cells in the wound microenvironment to express *Spp1* and initiate scarring. No subsequent studies have analyzed

[1]Department of Pathology, School of Medicine, Nagasaki University, Nagasaki 852-8523, Japan. [2]Laboratory of Human Immunology (Single Cell Genomics), WPI Immunology Research Center, Osaka University, Suita, Osaka 565-0871, Japan. [3]Department of Plastic and Reconstructive Surgery, School of Medicine, Keio University, Shinjuku-ku, Tokyo 160-8582, Japan. [4]Genome Information Research Center, Research Institute for Microbial Diseases, Osaka University, Suita, Osaka 565-0871, Japan. [5]Laboratory Animal Resource Center, Transborder Medical Research Center, University of Tsukuba, Tsukuba, Ibaraki 305-8575, Japan. [6]Department of Forensic Pathology and Science, School of Medicine, Nagasaki University, Nagasaki 852-8523, Japan. [7]Department of Forensic Medicine, Faculty of Medicine, Kagawa University, Kita, Kagawa 761-0793, Japan. [8]Department of Tumor and Diagnostic Pathology, Atomic Bomb Disease Institute, Nagasaki University, Nagasaki 852-8523, Japan. [9]Department of Biochemistry, Biomedical Sciences, University of Bristol, Bristol BS8 1TD, UK. [10]Department of Bioinformatic Engineering, Graduate School of Information Science and Technology, Osaka University, Suita, Osaka 565-0871, Japan. [11]Department of Tissue Repair and Regenerative Medical Science, Atomic Bomb Disease Institute, Nagasaki University, Nagasaki 852-8523, Japan. [12]Leading Medical Research Core Unit, Graduate School of Biomedical Sciences, Nagasaki University, Nagasaki 852-8523, Japan. [13]These authors contributed equally: Shigeto Seno, Daisuke Okuzaki. ✉E-mail: senoo@ist.osaka-u.ac.jp; dokuzaki@biken.osaka-u.ac.jp; ryoichi@nagasaki-u.ac.jp

skin wound healing in these mice at the single-cell level to reveal information about cell position and fate at wound sites.

Single-cell RNA sequencing (scRNA-Seq) technology can be used to comprehensively identify and quantify the transcriptome of individual cells and define unique cell types. However, this method lacks spatial information regarding the location of cells within tissues. Spatial transcriptome analysis using the 10× Genomics Visium platform can be used to define the spatial topography of gene expression in tissue microenvironments. Using this technique, the gene expression patterns and morphological architecture of organs (Asp et al, 2019; Maynard et al, 2021), diseases (Maniatis et al, 2019), and skin wound sites (Foster et al, 2021) have been identified in humans and mice. However, Visium spatial transcriptomics cannot be used to verify gene expression at the single-cell level, so an integrated method combining Visium and single-cell analyses is necessary. Foster and colleagues utilized scRNA-Seq data as an intermediary to integrate Visium and scATAC-Seq data; their approach primarily focused on fibroblasts while excluding the contributions of non-fibroblasts (Foster et al, 2021). Consequently, a novel integration platform is needed to deconvolve mixed cell-type populations within Visium spatial spots using scRNA-Seq datasets, thereby furthering integrated multiomics research.

Here, our scRNA-Seq analysis reveals that a minority population of wound fibroblasts expressing high levels of *Acta2* (*Acta*$^{high}$, a marker for myofibroblasts (McAndrews et al, 2022)) plays a crucial role in determining the fate of skin wound healing. Additionally, we establish super-resolved spatial transcriptomics by the integration of Visium and scRNA-Seq data, and using this approach, identify several inflammation-related scarring genes, including integrin beta-like 1 (*Itgbl1*) using *PU.1*$^{-/-}$ mice. *Itgbl1* supports collagen assembly at skin wound sites, and *Itgbl1* in myofibroblasts is important for granulation tissue formation. *Itgbl1* expression, myofibroblast differentiation, and fibrogenesis are regulated by the antagonism of TGFβ1 and IL1β signaling. Overall, these data extend our understanding of skin wound healing and provide tools for future research concerning inflammation and fibrosis.

# Results

## Temporal dynamic changes in single-cell populations and genes expressed during healing of murine skin wounds

Time-course scRNA-Seq datasets generated from murine skin wound tissues have the potential to provide insight into the functions and dynamics of molecularly defined cell types. We performed scRNA-Seq on cells from epidermal, dermal, granulation, and fascia tissues from wound sites of C57BL/6J mice on Days 3, 7, and 14 post injury using the 10× Genomics platform (Fig. 1A; Dataset EV1). First, we obtained an overview of the diversity of cell populations in these skin wounds by integrating the data from each tissue type using the Seurat batch-alignment algorithm to correct for the method-associated batch effect (Hao et al, 2021). Data analysis of the 40,024 cells that met our quality control criteria resulted in unbiased clustering and visualization with a uniform manifold approximation and projection (UMAP) plot displaying eight clusters (Becht et al, 2018) (Appendix Fig. S1). We then performed semi-automated cell-type annotation using a BioTuring Browser (BBrowser) database to confirm that all clusters

represented known constituents of murine skin wounds, and that each cluster had a unique gene expression pattern encompassing several of the following representative cell-specific markers: neutrophils (*S100a9*), macrophages (*Lyz2*), T cells, (*Cd3g*), fibroblasts (*Dcn*), epithelial cells (*Krt10*), endothelial cells (*Pecam1*), pericytes (*Rgs5*), and mesenchymal cells (*Bmp4*) (Appendix Fig. S1; Dataset EV2).

Next, we performed subclustering of these populations and found further heterogeneity and unique gene expression profiles, revealing the following: six subclusters of macrophages (M1 through M6); three subclusters of dendritic cells (D1 through D3) from the *Lyz2*-positive primary macrophage subcluster; eight subclusters of fibroblasts from the *Dcn*-positive primary fibroblast cluster (F1 through F8); and one subcluster each of B cells and Schwann cells (Fig. 1B; Appendix Fig. S2A; Dataset EV3). We found that the majority of skin wound healing-related cells comprised macrophages (15,566 cells, 39%) and fibroblasts (15,435 cells, 39%) (Fig. 1C). Some of the macrophage and fibroblast subclusters exhibited characteristic temporal patterns that presumably reflect changes in function during the repair process. The numbers of macrophages in M3, M4, and M6 markedly decreased on Day 14, while F4, F6, and F8 distinctively appeared on Days 7 and 14 post injury (Fig. 1B,C). Peak numbers of fibroblasts appeared on Day 7 (F2, F3, F6, F7, and F8) and Day 14 (F1, F4, and F5). Interestingly, excluding F5, the mRNA expression levels and numbers of expressed genes were markedly higher in fibroblast subclusters than in the other cell types (Fig. 1D; Appendix Fig. S2B). These results indicate that fibroblasts infiltrating wound sites are highly activated. Taken together, our scRNA-Seq datasets revealed temporal dynamic changes in single-cell populations throughout the course of wound healing, along with unique gene expression patterns that likely reflect key episodes in this process.

## Gene expression heterogeneity in macrophages involved in skin wound healing

Wound macrophages have diverse roles in inflammation and tissue repair. During wound healing, these macrophages switch between pro-inflammatory M1 and anti-inflammatory M2 phenotypes (Hassanshahi et al, 2022). To identify the characteristic gene markers and macrophage phenotype differentiating each macrophage subcluster, we analyzed gene expression patterns (Fig. EV1; Appendix Table S1). We found that the M1 subcluster exhibited high expression levels of various classically activated macrophage markers, including antigen processing/presentation genes (*Cd74*, MHC class II *H2-EB1* and *H2-AB1*), *Ccr2*, and *Il1b*. Additionally, the M6 subcluster specifically expressed inflammation-associated genes, including *S100A8/A9*, *G0s2*, *Slpi*, *Acod1*, *Hdc*, and *Il1r2*, suggesting that M1 and M6 subcluster macrophages could be classically polarized macrophages. In contrast, the M2 and M3 subclusters expressed alternatively activated polarized macrophage markers and anti-inflammatory cytokines: *Mrc1*, *F13a1*, *Wfdc17*, *Cbr2*, *Gas6* in M2, and *Arg1*, *Spp1*, *Ctsd*, *Cd36*, *Lgals3*, and *Fth1* in M3. The M5 subcluster macrophages specifically expressed genes associated with cell proliferation (*Stmn1*, *Hmgb2*, *Mki67*, and *Top2*), as well as the cytoskeleton regulatory gene *Tuba1b*. *Hmgb1*, a gene known to play a key role in the initiation of innate and adaptive immune responses, was also markedly expressed in

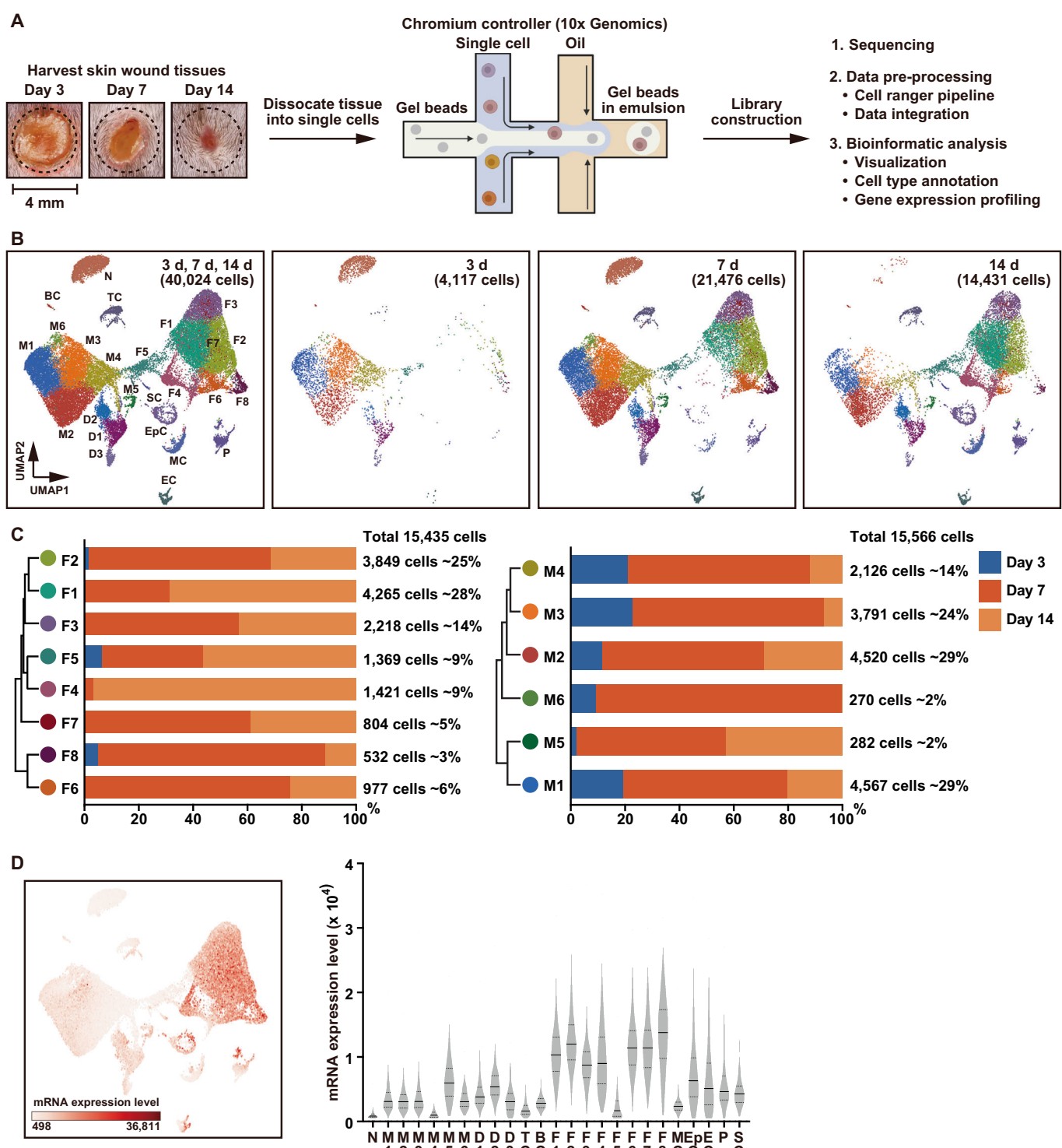

M5 subcluster macrophages, suggesting that wound-infiltrating macrophages proliferate in response to polarization toward the pro-M1 or pro-M2 phenotypes. Their proliferation could contribute to macrophage accumulation at skin wound sites (Pang et al, 2020). Compared with other subclusters, the M4 subcluster macrophages exhibited markedly low gene expression (Fig. 1D; Appendix Fig. S2B). Collectively, our findings indicate that the M5 subcluster comprised proliferation-associated macrophages, suggesting that

wound-infiltrating macrophages may proliferate before differentiating into polarized macrophages.

## Gene expression heterogeneity in fibroblasts involved in skin wound healing

Dermal fibroblasts are crucial for skin structure, supporting resident cells and forming granulation tissue, which, in turn, can

Figure 1. Dynamic temporal changes in single-cell-derived populations and gene expression at skin wound sites in mice.

(A) Overview of single-cell analysis. (B) UMAP plot of integrated data from 40,024 cells collected on Days 3 (3 d; 4117 cells), 7 (7 d; 21,476 cells), and 14 (14 d; 14,431 cells) post skin wounding, showing each cell type. N neutrophils, BC B cells, TC T cells, M macrophages, D dendritic cells, F fibroblasts, SC Schwann cells, EpC epithelial cells, MC mesenchymal cells, P pericytes, EC endothelial cells. (C) Bar plots of unsupervised hierarchical clustering showing relatedness of wound fibroblast and macrophage subclusters, and proportions of fibroblast and macrophage subclusters on Days 3 (blue), 7 (red), and 14 (orange). (D) UMAP of mRNA expression levels across 40,024 cells at Days 3, 7, and 14 (left panel), and a violin plot showing total mRNA expression levels in the various subclusters (right panel). N: $n = 2567$; M1: $n = 4567$; M2: $n = 4520$; M3: $n = 3791$; M4: $n = 2126$; M5: $n = 282$; M6: $n = 270$; D1: $n = 1,359$; D2: $n = 734$; D3: $n = 467$; TC: $n = 744$; BC: $n = 122$; F1: $n = 4265$; F2: $n = 3849$; F3: $n = 2218$; F4: $n = 1421$; F5: $n = 1369$; F6: $n = 977$; F7: $n = 804$; F8: $n = 532$; MC: $n = 829$; EpC: $n = 854$; EC: $n = 507$; P: $n = 732$; SC: $n = 118$. Source data are available online for this figure.

lead to scarring at the wound site (Peña and Martin, 2024). We analyzed gene expression patterns of the fibroblast subclusters and compared them with the patterns displayed by all cell-type clusters (Fig. EV2A; Appendix Table S2). Fibroblast subclusters associated with granulation tissue formation (F2, F3, F6, F7, and F8) exhibited specific gene expression patterns that were consistent with various cell types. The F2 subcluster expressed genes necessary for skin wound healing (*Postn*, *Col5a3*, *Lgals1*, and *Vcan*), the collagen fibrogenesis gene *Plod2*, and the fibrosis-related gene *Loxl2*. The F3 subcluster expressed the pan-fibroblast marker *Pi16*, as well as genes that have been identified in activated fibroblasts (*C3*, *Prss23*, *Cd55*, *Sema3c*, *Anxa3*, *Dpp4*, and *Scara5*). The F6 subcluster expressed several myofibroblast markers (*Acta2* and *Tagln*) and myofibroblast functional genes (*Col12a1*, *Lrrc15*, *Tpm1*, and *Cdh11*), as well as ECM remodeling genes (*Thbs2* and *Mmp14*). Although the F7 subcluster was recognized as fibroblastic, it also expressed many macrophage-related genes (*Cd74*, *Lyz2*, *Fcer1g*, *Apoe*, *C1qc*, *Tyrobp*, and *C1qa*), suggesting that it might comprise myeloid-derived fibroblasts. The F8 subcluster expressed genes related to proliferation (*Mki67* and *Pclaf*), cell cycle (*Stmn1* and *Cks2*), and chromatin/cytoskeleton modulation (*Tubb5*, *Hmgb2*, *Hmgb1*, *H2afz*, *Birc5*, and *Nucks1*). Interestingly, these cells also showed marked expression of two myofibroblast markers (*Acta2* (McAndrews et al, 2022) and *Tagln* (Lawson et al, 1997)), along with the inflammation-related scarring gene *Spp1* (Mori et al, 2008), suggesting a direct correlation between the F8 and F6 subclusters in myofibroblast differentiation on Day 7 at skin wound sites (Fig. EV2B).

The number of "remodeling" fibroblasts, which play a key role in ECM deposition and remodeling during scar formation, peaked at Day 14 (F1, F4, and F5 subclusters) and exhibited characteristic gene expression patterns. The F1 subcluster showed high expression of various ECM component/fibrosis genes (*Mgp*, *Eln*, *Cilp*, *Igf1*, *Cygb*, and *Gas6*), the fibroblast-to-myofibroblast transition gene *Smoc2*, and the homeostasis-related gene *Mmp3*. The F4 subcluster showed significantly increased expression of ECM components (*Mfap4*, *Pcolce2*, *Ecm2*, and *Col15a1*), the ECM homeostasis gene *Ctsk*, and the reticular fibroblast marker *Ppp1r14a*, compared with the other subclusters. Gene expression in the F5 subcluster was markedly low compared with that in the other subclusters (Fig. 1D; Appendix Fig. S2B).

## Critical role of minority *Acta2*[high] myofibroblasts in the fate of skin wound healing

Myofibroblasts play key roles in tissue repair by producing ECM, promoting wound contraction, and secreting growth factors. They interact with other cells in wound sites and are crucial for effective

repair (McAndrews et al, 2022). We identified three unique minority subpopulations of fibroblasts (F6 and F8 subclusters at Day 7 and F4 subcluster at Day 14) that appeared during skin wound healing (Fig. 1B,C). Interestingly, two markers for myofibroblasts, *Acta2* and *Tagln*, were identified in the F6 and F8 subclusters (Fig. EV2A,B). Furthermore, *Acta2*[high] fibroblasts in the F8 subcluster simultaneously expressed genes related to proliferation (*Mki67* and *Pclaf*), cell cycle (*Stmn1* and *Cks2*), and chromatin/cytoskeleton modulation (*Tubb5*, *Hmgb2*, *Hmgb1*, *H2afz*, *Birc5*, and *Nucks1*) (Fig. EV2A), suggesting that the F8 subcluster may have been a source of F6 subcluster fibroblasts. Although the numbers of cells in the F6 and F8 subclusters were markedly low compared with those in the other fibroblast subpopulations, their expression profiles suggest that they are myofibroblasts and that they play critical roles in tissue repair.

## Characteristic cell–cell interactions of minority fibroblasts using CellChat

Wound-infiltrating cells regulate the inflammatory response and tissue repair through the secretion of bioactive substances and cell–cell communication at skin wound sites. To better understand the global communication patterns among these cells, we sought to accurately identify the signaling links associated with each cell subcluster and to perform effective systems-level analyses of those links. We investigated cell–cell communication using CellChat, a platform that infers, visualizes, and analyzes intercellular communication from scRNA-Seq data (Jin et al, 2021) (Figs. EV3–5). We found associations mainly with M1, M2, and M3 macrophage subclusters, as well as the F8 fibroblast cluster (highly likely to undergo phenotypic transformation into myofibroblasts on Day 7 post injury), were present in the crosstalk between macrophages and fibroblasts during the early stages of wound healing (Day 3 post injury) (Fig. EV3B–F). We speculate that TGFβ1 (mainly associated with M1 and M3), CCL8 (mainly associated with M2), SPP1 (mainly associated with M3), and other genes may influence activation of the F8 subcluster (Fig. EV3B–F).

On Day 7 post injury (Fig. EV4B–F), thrombospondin (THBS) (M1, M3, and F2), transforming growth factor β (TGFB) (M1, M2, M3, and F2), Visfatin (M1, M2 and M3), growth differentiation factor (GDF) (M2 and M3), SPP1 (M2 and M3), and oncostatin M (OSM) (M1 and M3) interactions appeared to be the predominant contributors to F6 and F8 subcluster activation. By contrast, fibronectin (FN) 1, angiopoietin-like protein (ANGPTL), Periostin, epidermal growth factor (EGF), and non-canonical WNT (ncWNT) signaling directly activated the F6 and F8 subclusters.

On Day 14 post injury (Fig. EV5B–F), we performed CellChat analysis focusing on subclusters F1 and F4, which were

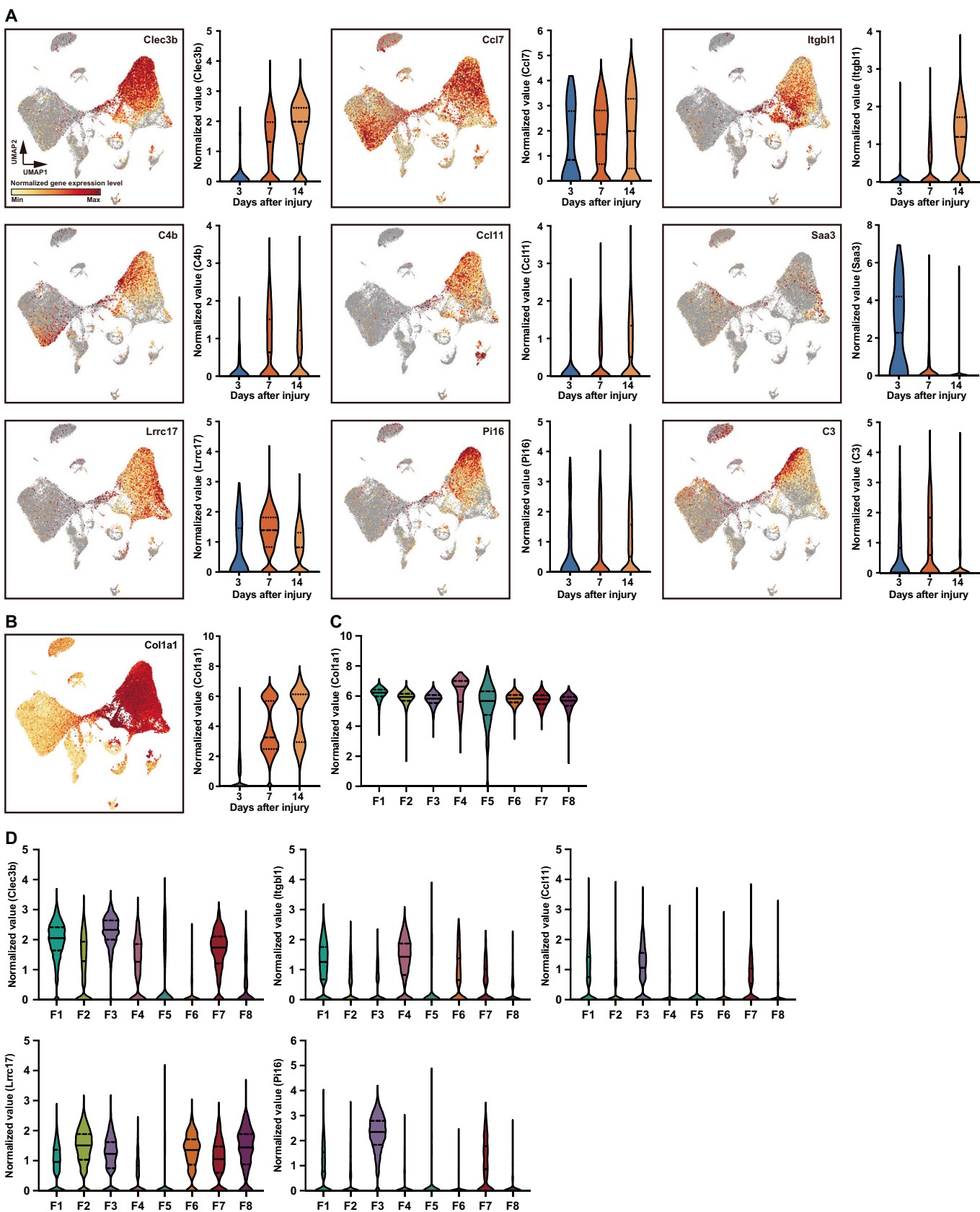

◀ **Figure 2. Expression of inflammation-related gene candidates in the fibroblast subclusters.**

(A) UMAP plots (left) and violin plots (right) of normalized expression levels of the indicated genes from integrated data collected on Days 3 ($n = 4117$), 7 ($n = 21,476$), and 14 ($n = 14,431$) post skin wounding. (B) UMAP plot (left) and violin plot (right) of *Col1a1* expression levels from integrated data collected on Days 3 ($n = 4117$), 7 ($n = 21,476$), and 14 ($n = 14,431$) post skin wounding. (C) Normalized *Col1a1* expression level in each fibroblast subcluster (F1–F8). F1: $n = 4265$; F2: $n = 3849$; F3: $n = 2218$; F4: $n = 1421$; F5: $n = 1369$; F6: $n = 977$; F7: $n = 804$. (D) Normalized expression levels of *Clec3b*, *Itgbl1*, *Ccl11*, *Lrrc17*, and *Pi16* in F1 to F8 from integrated data collected on Days 3, 7, and 14 post skin wounding. F1: $n = 4265$; F2: $n = 3849$; F3: $n = 2218$; F4: $n = 1421$; F5: $n = 1369$; F6: $n = 977$; F7: $n = 804$. Source data are available online for this figure.

characteristically distinct. We found that each subcluster was markedly activated by itself through ECM-related signaling (i.e., FN1, THBS, and Periostin). Subclusters F1 and F4 interacted with subclusters M1 and M2, as indicated by marked signaling through GDF, TGFB, Visfatin, and OSM. Interestingly, only the F4 subcluster communicated with inflammatory cells through interleukin (IL)1 signaling (Appendix Fig. S3A). These results revealed that macrophage and fibroblast subclusters work cooperatively to facilitate intercellular communication, including the aforementioned signals.

## Identification of inflammation-related genes in murine skin wound healing

To identify inflammation-related genes and fibroblast-derived genes that are associated with scarring during skin wound healing in mice, we made incisional wounds in neonate littermates of WT versus $PU.1^{-/-}$ mice, which lack an inflammatory response (Martin et al, 2003). We performed bulk mRNA sequencing and identified 1057 candidate inflammation-related genes that had downregulated expression at wound sites of $PU.1^{-/-}$ mice compared with WT mice (Dataset EV4). To further screen for inflammation-related "scarring" genes expressed by wound-infiltrating fibroblasts, we compared the results of scRNA-Seq with the 1,057 candidate gene sequences and identified five genes in subcluster F1, four genes in subcluster F2, nine genes in subcluster F3, one gene in subcluster F4, two genes in subcluster F6, 15 genes in subcluster F7, and two genes in subcluster F8 (Dataset EV5). A comparison of these genes with scRNA-Seq data (Dataset EV3) identified *Clec3b* (F1, F3, F7), *Ccl7* (F1, F2, F3, F7), *Itgbl1* (F1, F4), *C4b* (F1, F3), *Ccl11* (F1, F3), *Saa3* (F2, F8), *Lrrc17* (F2, F6, F8), *Pi16* (F3, F7), and *C3* (F3, F7) as potential inflammation-related genes that are expressed by wound-infiltrating fibroblasts.

To identify fibroblast-specific genes, we investigated the temporal expression patterns of the inflammation-related genes using our scRNA-Seq data. We found that *Clec3b*, *Itgbl1*, *Ccl11*, *Lrrc17*, and *Pi16* were specifically expressed by wound-infiltrating fibroblasts (Fig. 2A). Collagen is the most abundant skin ECM protein and its excessive and aberrant deposition by fibroblasts at wound sites is associated with scarring. Therefore, we hypothesized that a subpopulation of fibroblasts expressing collagen might be pivotal in driving scarring. Indeed, expression of the type I collagen gene (*Col1a1*) was significantly increased in fibroblast subclusters at Days 7 and 14 post injury compared with Day 3 (Fig. 2B). Furthermore, subcluster F4 showed much higher expression of *Col1a1* than the other fibroblast subclusters (Fig. 2C), suggesting that these cells might be most strongly associated with scar formation. We further analyzed the expression patterns of *Clec3b*, *Itgbl1*, *Ccl11*, *Lrrc17*, and *Pi16* in the fibroblast subclusters and found that *Itgbl1* was characteristically expressed by F4 and F1

fibroblasts (Fig. 2D). We also observed that the next-highest level of *Col1a1* expression occurred in the F1 subpopulation. Because the appearance of the F4 and F1 subclusters was marked in the maturation phase (Day 14), we suspected that *Itgbl1* might be associated with scarring.

## Transcriptional spatial gene expression analysis in murine skin wound healing

To comprehensively characterize spatial transcriptional expression of inflammation-related genes, we used the Visium platform (Fig. 3A). Our analysis included up to 697 under-tissue spots, which are data points considered equivalent to microdissections containing approximately 20–30 cells per spot, with a median of up to 2936 expressed genes per spot (Dataset EV6). We tested the expression of nine candidate inflammation-related genes on the Visium platform (Fig. 3B) and confirmed that their expression patterns were similar to the results of single-cell analysis.

Next, we performed differentially expressed gene (DEG) analysis in wound regions that were manually annotated, as well as in each cluster of tissue (Appendix Fig. S4; Datasets EV7–EV9). We observed that almost all of the significantly highly expressed genes were extracted in similar clusters by both manual and automatic annotation. Therefore, we used the results of manually annotated gene set analysis for further experiments. When we compared Visium transcriptomics data regarding genes with significantly upregulated expression in wound sites at Days 3, 7, and 14 (Datasets EV7–EV9) and the 1,057 candidate inflammation-related genes (Dataset EV4), we found that 19 genes from the Days 3 and 7 wound sites, and nine genes from the Day 14 wound sites, could be classified as candidate inflammation-related genes (Dataset EV10). This analysis successfully identified *Spp1*, a previously reported inflammation-related scarring gene (Mori et al, 2008), suggesting that our combination of Visium transcriptomics and $PU.1^{-/-}$ mice analyses was effective in the identification of inflammation-related scarring genes. Additionally, our scRNA-Seq analysis identified *Clec3b*, *Itgbl1*, *Saa3*, *Lrrc17*, *Pi16*, and *C3* as potential scarring-associated genes. Taken together, these results demonstrated that the combination of scRNA-Seq, Visium transcriptomics, and $PU.1^{-/-}$ mouse data was a useful approach for screening candidate genes involved in inflammation and scarring during skin wound healing.

## Establishment of super-resolved spatial transcriptomics by integration with Visium and scRNA-Seq

In Visium analysis, multiple cells are mapped into a single capture area with a circular diameter of 55 μm, depending on the tissue and spatial location (Fig. 3A). However, because our subject was a single cell, the observed spatial expression profile had to be separated into

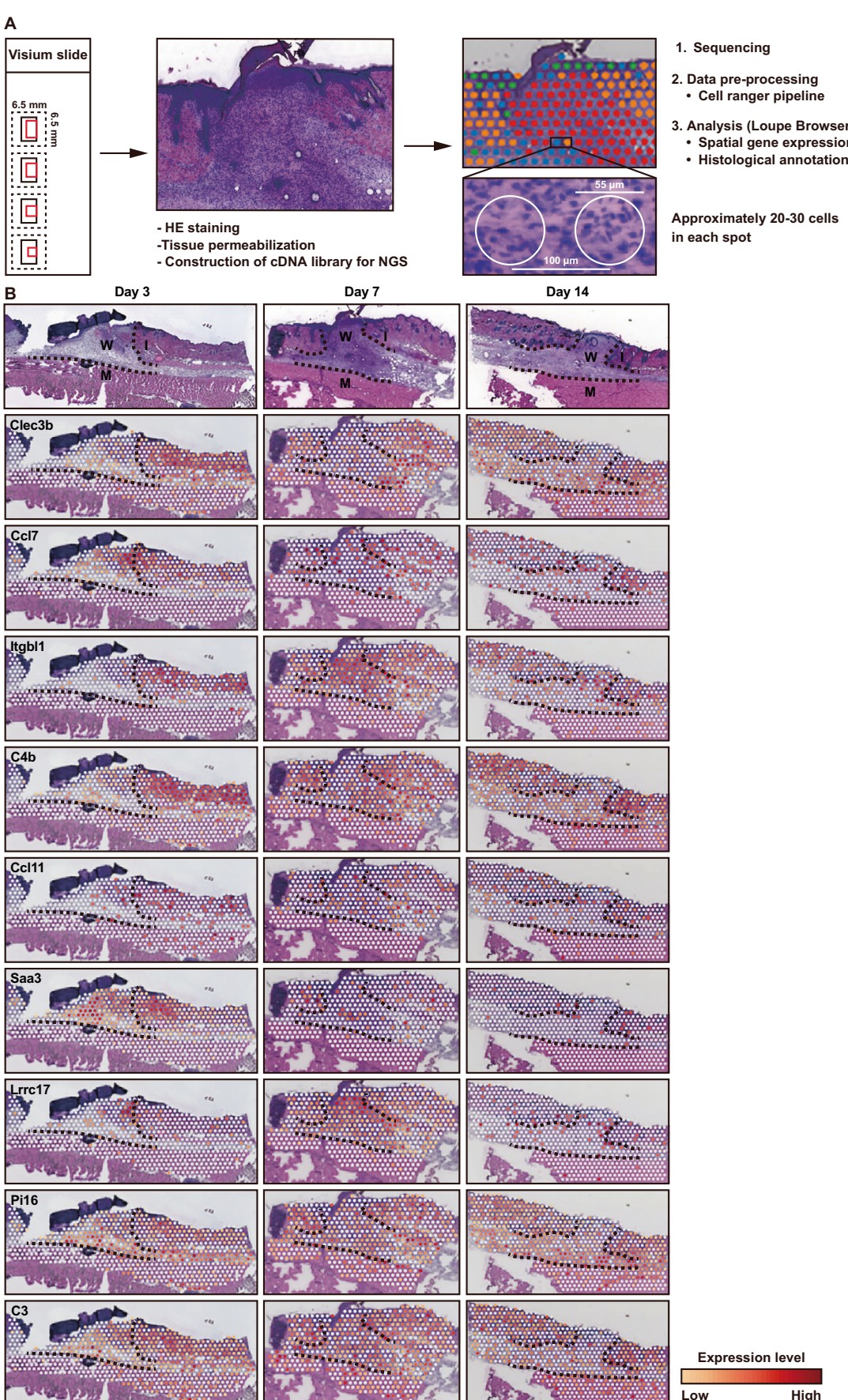

**Figure 3. Spatial transcriptome analysis of inflammation-related genes using the Visium platform.**

(A) Flowchart of the Visium process. (B) Spatial expression of the indicated inflammation-related genes at skin wound sites on Days 3, 7, and 14 post wounding. W wound site, I intact skin, M muscle. H&E staining images H&E staining images in Figs. 3B and 4B–D were obtained using single sets of Visium data (Appendix Fig. S4B,G,L). Source data are available online for this figure.

signals from individual cells using a process called spatialomics deconvolution (Cable et al, 2022). To further advance the resolution of Visium data, we integrated scRNA-Seq and Visium data using the robust cell-type decomposition (RCTD) method (Cable et al, 2022), then performed spatialomics deconvolution to comprehensively map the tissue architecture (Fig. 4A). At Day 3 post injury, neutrophils were observed underneath the scab layer (Fig. 4B). By Day 7 post injury, the ratio of macrophages to fibroblasts had increased, whereas the number of neutrophils had markedly decreased (Fig. 4C). Additionally, most of the spots where macrophages and fibroblasts had been observed previously remained consistent, suggesting an interaction between these cell types within the wound microenvironment. The detection of epithelial cells at sites of re-epithelialization was confirmed, as was the high ratio of endothelial cells to other cell types at sites of neovascularization. At Day 14 post wound injury, the ratio of fibroblasts to other cell types was highest, followed by the ratio of macrophages to other cell types (Fig. 4D). Although it can be challenging to precisely identify cell types using Visium analysis, knowledge of specific cell markers (Fig. 4B,C) and spatialomics deconvolution analysis enabled us to overcome this limitation.

Our CellChat analysis revealed that major signaling pathways had the greatest impact on outgoing or incoming signaling for specific fibroblast and macrophage subpopulations (Figs. EV3–5). Signaling directions have been shown to play crucial roles in organizing human skin and murine brain tissue, as demonstrated using the communication analysis by optimal transport (COMMOT) method (Cang et al, 2023). This method considers competition between different ligand and receptor species, as well as spatial distances between cells. To further elucidate wound macrophage-to-fibroblast communication in our super-resolved spatial transcriptomics studies, we performed COMMOT analysis which showed that TGFβ signaling was directed toward the center of the granulation tissue on Day 7 after wounding. Conversely, ECM-related signaling (SPP1, FN1, THBS, and PERIOSTIN) was directed from the center of the granulation tissue to the wound edge (Fig. 4E). We observed a similar trend on Day 14 post wounding (Fig. 4F). These results suggest that the strength of bioactive substance signaling influences the center of the wound, whereas the strength of ECM interaction signaling influences the transition from wound center to the margins of the wound. These consistent signaling directions likely contribute to an effective wound-repair process.

## Delayed skin wound healing and altered collagen fibril formation at scar sites *Itgbl1*$^{-/-}$ mice

*Itgbl1* affects ECM deposition and is involved in hepatitis B virus-induced liver fibrosis (Wang et al, 2017). Recently, Zhao and colleagues reported that reduced *ITGBL1* expression levels were strongly associated with the anti-scarring effects of pressure therapy in clinical specimens (Zhao et al, 2023). Of all inflammation-related genes identified in the present study, *Itgbl1* showed the strongest expression in subclusters F4 and F1, which

appeared late in the repair process (on Day 14 post injury), suggesting that *Itgbl1* plays a functional role in scarring at skin wound sites. Therefore, we investigated the molecular mechanisms of *Itgbl1* in skin wound healing.

First, we examined the expression of *Itgbl1* using quantitative PCR (qPCR) and found that it was significantly increased on Days 7, 10, and 14 post injury compared with intact skin, supporting our scRNA-Seq and Visium data (Fig. 5A). In situ hybridization (ISH) revealed that wound-infiltrating spindle-shaped cells, presumed to be fibroblasts, expressed *Itgbl1* at skin wound sites on Day 7 post injury (Fig. 5B–D). These results indicate that *Itgbl1* may indeed be involved in granulation tissue formation.

To investigate the function of *Itgbl1* in vivo, we generated full-body *Itgbl1*$^{-/-}$ mice using CRISPR-Cas technology as previously described (Tanimoto et al, 2022) (Fig. 5E). We confirmed the absence of phenotypic abnormalities in 8-week-old full-body *Itgbl1*$^{-/-}$ mice. The lack of *Itgbl1* mRNA expression at skin wound sites confirmed the *Itgbl1*$^{-/-}$ genotype (Fig. 5F). Our initial examination of the gross appearance of aseptic healing revealed a significant delay in *Itgbl1*$^{-/-}$ mice ($65.8\% \pm 21$) compared with that in WT mice ($35.2\% \pm 16$; $P < 0.0001$) on Day 7 post injury (Fig. 5G,H). This delayed healing was accompanied by increased granulation tissue area on Days 7 and 14 post injury (Fig. 5I,J).

Next, to investigate the potential effect of altered *Itgbl1* expression on the development of scarring, we examined gross collagen fibril formation at wound sites using picrosirius red staining and found that collagen bundle organization was markedly reduced in *Itgbl1*$^{-/-}$ mice at Day 14 post injury (Fig. 5K). To further analyze the development of scarring, we performed transmission electron microscopy (TEM). Interestingly, the fibril diameter at the mid-region of wound sites was markedly reduced in *Itgbl1*$^{-/-}$ mice ($42.7 \pm 6.54$ nm) compared with that in WT mice ($50.0 \pm 14.2$ nm; $P < 0.0001$) (Fig. 5L). We confirmed that hydroxyproline content at scar sites was significantly decreased in *Itgbl1*$^{-/-}$ mice ($3.91 \pm 0.93$ mol/L) compared with WT mice ($4.99 \pm 0.84$ mol/L; $P < 0.01$) (Fig. 5M). This difference occurred despite the lack of differences in *Tgfb1*, *Acta2*, *Col1a1*, and *Col3a1* mRNA expression, or the *Col1a1/Col3a1* ratio, at wound sites in either mouse group on Days 7 and 14 post injury (Appendix Fig. S5). By contrast, there were no significant differences in collagen fibril morphology within intact skin from WT ($104.7 \pm 30.4$ nm) and *Itgbl1*$^{-/-}$ ($107.4 \pm 39.1$ nm) mice (Appendix Fig. S6A). We speculate that these differences in collagen assembly at wound sites play a key role in the reduced scar formation observed during the maturation phase of healing in *Itgbl1*$^{-/-}$ mice.

## *Itgbl1* expression, myofibroblast differentiation, and fibrogenesis regulated by antagonism of TGFβ1 and IL1β signaling

The TGFβ1 signaling pathway plays a key role in scarring by promoting the transition of fibroblasts into myofibroblasts, which

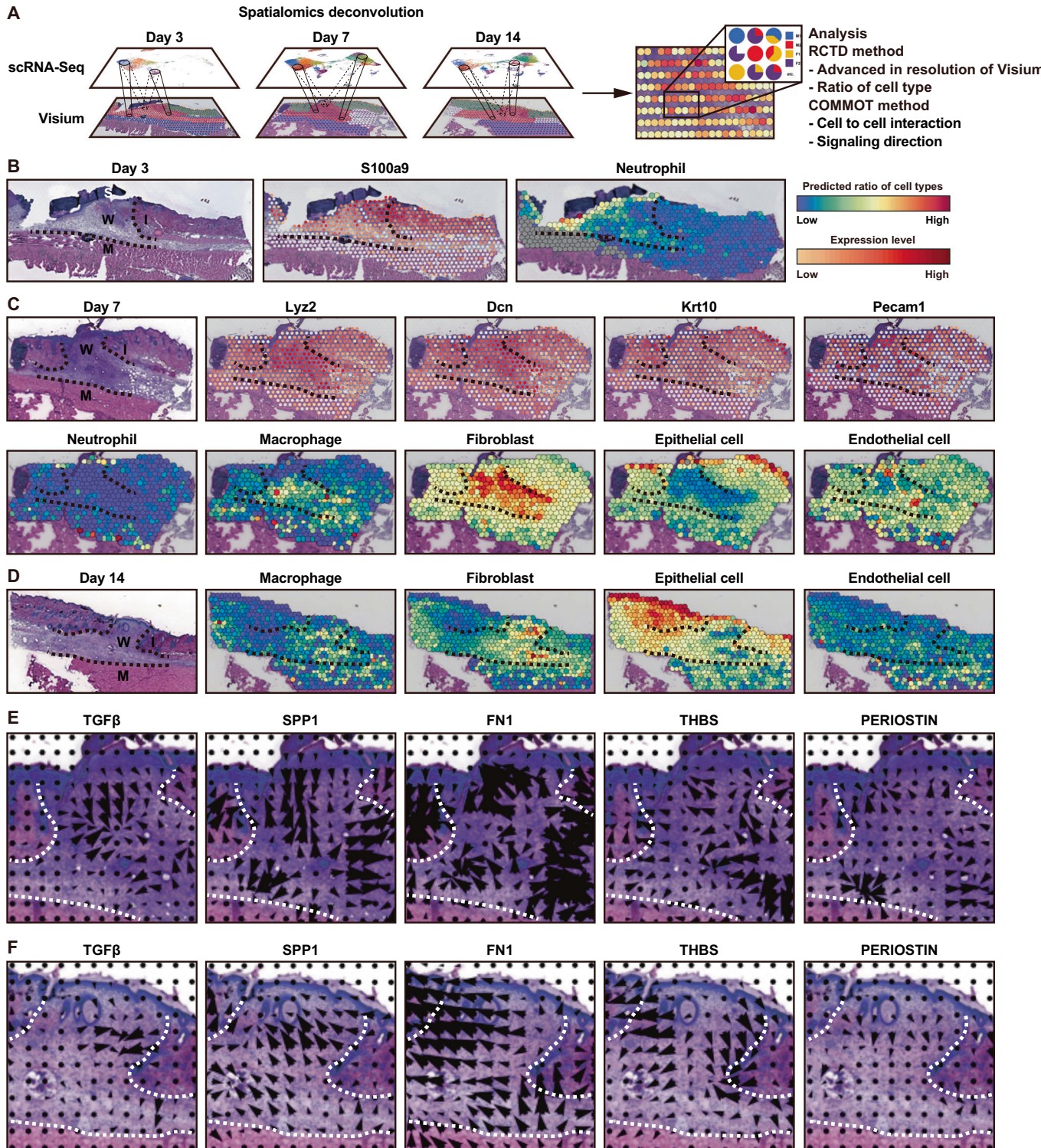

**Figure 4. Cell location, cell–cell communication, and signaling analysis of spatial transcriptomics data with spatialomics deconvolution.**

(**A**) Overview of the spatialomics deconvolution process. (**B–D**) Identification of cell location in spatial transcriptomics data with spatialomics deconvolution at skin wound sites on Days 3 (**B**), 7 (**C**), and 14 (**D**) post wounding. Expression of *S100a9* (neutrophil marker), *Lyz2* (macrophage marker), *Dcn* (fibroblast marker), *Krt10* (epithelial cell marker), and *Pecam1* (endothelial cell marker) from Visium. The ratio of cell types is predicted by spatial deconvolution. The color bars in each panel represent relative values, with the range varying from panel to panel. S scab, W wound site, I intact skin, M muscle. H&E staining images in Figs. 3B and 4B–D were obtained using single sets of Visium data (Appendix Fig. S4B,G,L). (**E, F**) The signaling directions of five major signaling pathways using COMMOT on Days 7 (**E**) and 14 (**F**) post wounding. Source data are available online for this figure.

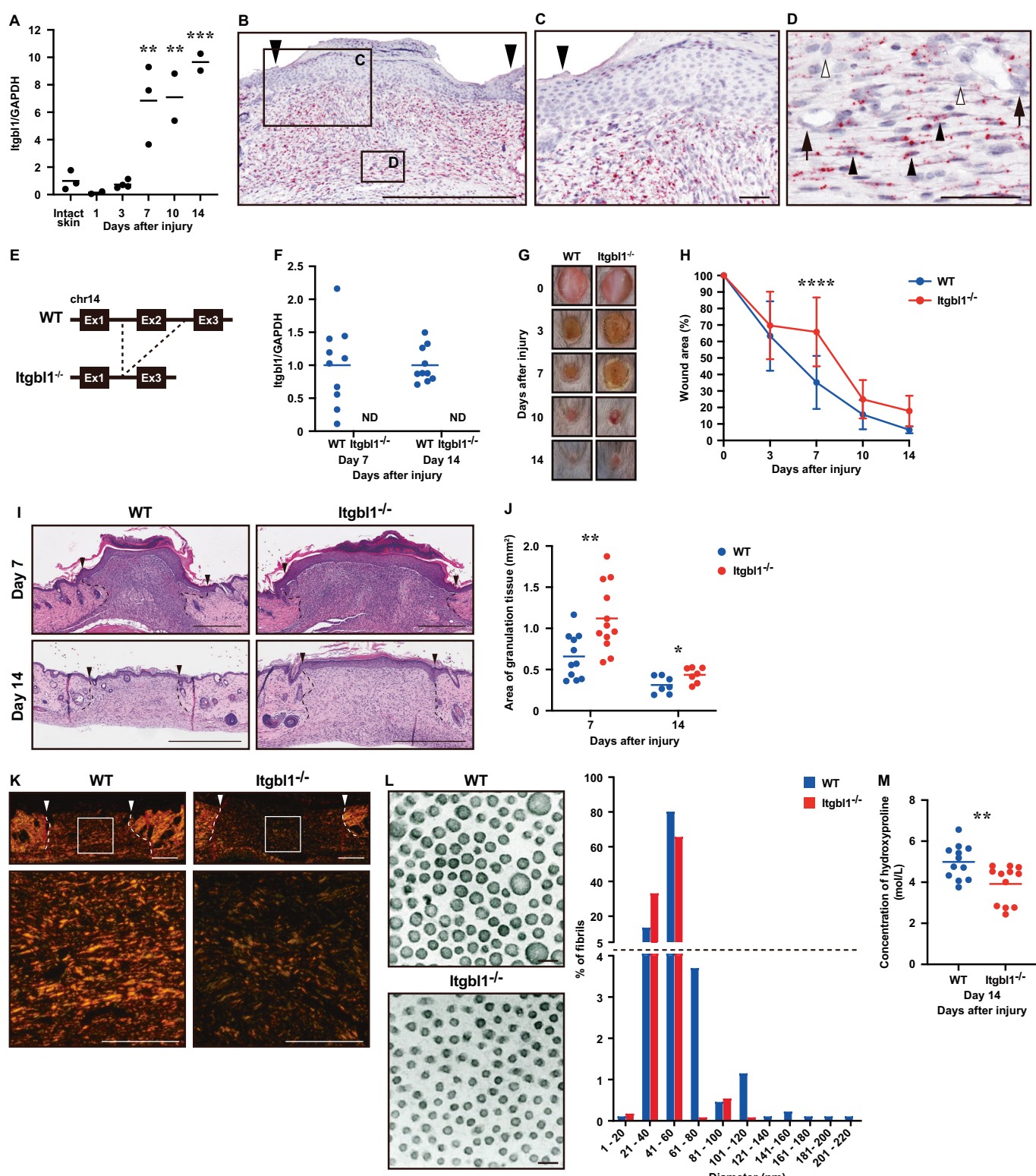

express the characteristic marker genes *Acta2* and *Tagln* (Lawson et al, 1997). Reportedly, TGFβ1-stimulation of neonatal rat cardiac fibroblast leads to the expression of integrin subunit β-like (ITGBL) 1, which is involved in myocardial fibrosis (Zhu et al, 2022). We tested the expression levels of *Itgbl1*, *Col1a1*, and *Acta2* in murine

dermal skin-derived primary fibroblasts (MDF) and human dermal skin-derived primary fibroblasts (HDF) and confirmed that these molecules were induced by TGFβ1-stimulation as shown in previous reports (Fig. 6A; Appendix Fig. S7A–C). Interestingly, human keloid-derived dermal fibroblasts constitutively expressed

**Figure 5. Delayed skin wound healing and altered collagen fibril formation at scar sites of the *Itgbl1* defect in mice.**

(A) qPCR measurement of the temporal expression of murine *Itgbl1* relative to that of *GAPDH* during skin wound healing ($n = 2$–4 wounds). *P* values: Intact skin vs. Day 7 = 0.0043, Intact skin vs. Day 10 = 0.0068, Intact skin vs. Day 7 = 0.0005. (B–D). ISH of *Itgbl1* at murine wound sites on Day 7 post injury. The area indicated with a rectangle in (B) is shown at higher magnifications in (C, D). Many *Itgbl1*-positive cells can be observed in granulation tissue, but not in the epidermis, which corresponded to the result from Visium analysis. Black arrowheads indicate the wound margin. (D) Expression of *Itgbl1* predominantly observed in wound-infiltrating spindle-shaped cells presumed to be fibroblasts (black arrowheads), but not in endothelial cells (black arrows) or macrophages (white arrowheads). Scale bars; (B) 500 μm, (C, D) 50 μm. (E) Schematic diagram of the generation of *Itgbl1*$^{-/-}$ mice illustrating the *Itgbl1* genomic construct containing a deletion of exon (Ex) 2. (F) qPCR analysis of *Itgbl1* in wound sites confirming that *Itgbl1*$^{-/-}$ mice were completely deficient in expression of *Itgbl1* (WT and *Itgbl1*$^{-/-}$, $n = 10$). ND none detected. (G) Representative photographic images of the gross appearance of excisional wounds in WT (left) and *Itgbl1*$^{-/-}$ (right) mice. (H) Proportion of the wound area remaining open at each time point relative to the initial wound area (WT; $n = 11$ wounds, *Itgbl1*$^{-/-}$; $n = 12$ wounds). *P* values: WT Day 7 vs. *Itgbl1*$^{-/-}$ Day 7 < 0.0001. (I) H&E staining of wound site on Days 7 and 14 post injury (wound margin [arrowheads]). Scale bars: 500 μm. (J) Measurement of granulation tissue area on Days 7 and 14 post injury in WT (Day 7; $n = 11$, Day 14; $n = 7$) and *Itgbl1*$^{-/-}$ mice (Day 7; $n = 12$, Day 14; $n = 7$). *P* values: Day 7 = 0.0048, Day 14 = 0.0391. (K) Polarized light microscopy differential interference contrast images of picrosirius red-stained sections of excisional wound sites on Day 14 post injury for analysis of collagen fibers and alignment. Arrowheads indicate the wound edge. Images are representative of eight independent experiments. Low-magnification images (upper) and high magnification of boxed areas (lower) are shown. Scale bars; (top) 200 μm, (bottom) 100 μm. (L) TEM images of collagen fibrils in connective tissue from mid-wound sites on Day 14 post injury (left). Histogram of the total range of fibril diameters in the wound site on Day 14 post injury (right); $n = 863$ fibrils from four WT mice; $n = 1092$ fibrils from four *Itgbl1*$^{-/-}$ mice. Scale bars: 100 nm. (M) Hydroxyproline content at skin wound sites on Day 14 post injury in WT and *Itgbl1*$^{-/-}$ mice ($n = 12$ wounds). *P* value: WT vs. *Itgbl1*$^{-/-}$ = 0.0069. Data information: All values represent the mean ± SD. One-way ANOVA followed by Dunnett's multiple comparisons test (intact skin vs. wound samples) (A), two-way ANOVA followed by Sidak's multiple comparisons test (H), and unpaired *t*-tests (J, M) were used to generate the indicated *P* values; *$P < 0.05$, **$P < 0.01$, ***$P < 0.001$, ****$P < 0.0001$.

*ITGBL1* (Fig. 6B), suggesting that *ITGBL1* could be involved in the exacerbation of scarring.

Our CellChat analysis revealed that only the F4 fibroblast subcluster, which specifically appeared on Day 14 post injury, received IL1β signaling from immune cells. This interaction was indicated by its expression of IL1 receptor type I (*Il1r1*) and IL1R receptor accessory protein (*Il1rap*) (Appendix Fig. S3A–C), and was confirmed by the low levels of IL1β mRNA and protein (13.6 ± 2.21 pg/μg) at wound sites on Day 14 post injury (Appendix Fig. S3D–F). We hypothesize that IL1β signaling could regulate F4 fibroblast subcluster activation and ECM production. To investigate IL1β signaling in fibroblasts, we treated MDF with IL1β and found that IL1β signaling significantly inhibited expression of *Col1a1* and *Acta2* in a dose dose-dependent fashion compared with the untreated control, although *Itgbl1* expression was not altered (Fig. 6C). Furthermore, *Itgbl1*, *Acta2*, *Col1a1*, and *Col3a1* expression in TGFβ1-stimulated MDF was markedly attenuated by IL1β (Fig. 6D). We suspect that IL1β may mediate the transformation of *Acta2*$^{high}$ myofibroblasts into *Acta2*$^{low}$ and/or *Acta2*-negative fibroblasts during late-stage healing.

Next, we confirmed the effects of IL1β signaling using HDF (Fig. 6E). Interestingly, a low concentration of IL1β (0.01 to 10 pg/mL) induced *ITGBL1*, whereas a high concentration of IL1β (100 pg/mL) suppressed *ITGBL1*. IL1β signaling markedly inhibited the expression of *COL1A1* and *ACTA2* in HDF, again in a dose-dependent fashion compared with untreated controls. Furthermore, *ITGBL1*, *ACTA2*, *Col1A1*, and *COL3A1* expression levels in TGFβ1-stimulated HDF were markedly attenuated by IL1β (Fig. 6F). Overall, IL1β expression was important for transforming myofibroblasts into other fibroblasts (Fig. 6G). Taken together, these results indicate that IL1β signaling acts to stabilize fibroblasts during the maturation phase by inhibiting myofibroblast activation.

## *Itgbl1* expression by myofibroblasts is important for granulation tissue formation

To visualize *Itgbl1*-expressing cells in vivo, we generated homozygous *Itgbl1-tdTomato* knock-in (*Itgbl1*$^{Tom/Tom}$) mice using CRISPR-Cas technology as previously described (Tanimoto et al,

2022) (Fig. 7A). Compared with intact skin, each wound site showed a marked increase in *Itgbl1*-expressing cells on Days 7 and 14 post injury (Fig. 7B). To investigate whether myofibroblasts express *Itgbl1*, we performed immunohistochemistry (IHC) for ACTA2 using *Itgbl1*$^{Tom/Tom}$ mice; the results showed that only 18% of cells present at wound sites on Day 7 post injury were double-positive for *Itgbl1* and ACTA2 (Fig. 7C), consistent with our single-cell analysis. To further confirm the presence of *Itgbl1*-expressing cells in vivo, we crossed *Itgbl1*$^{Tom/Tom}$ mice with Lysozyme M (*LysM*)-enhanced green fluorescent protein (EGFP) mice (Faust et al, 2000), in which most myelomonocytic cells specifically express EGFP (*Itgbl1*$^{Tom/Tom}$::*LysM*$^{EGFP/EGFP}$). We observed that *Itgbl1* was not expressed by wound-infiltrating macrophages on Day 7 post injury (Fig. 7D). Interestingly, most *Itgbl1*-expressing cells (possibly fibroblasts) were located close to macrophages, suggesting that *Itgbl1* expression in fibroblasts might be induced by bioactive substances derived from macrophages in granulation tissue on Day 7 post injury.

Cells expressing *Itgbl1* were also found in intact skin, suggesting that they might contribute to repair (Fig. 7B). To investigate the possible involvement of *Itgbl1* expression in skin healing, we generated *Itgbl1*$^{CreERT2/CreERT2}$ mice, crossed them with green-to-red fluorescent-convertible Cre-reporter mice (*ROSA26*$^{GRR/GRR}$) (Hasegawa et al, 2013), and examined the fate of *Itgbl1*-expressing cells in intact skin after tamoxifen-dependent recombination (Fig. 7E). *Itgbl1*-positive cells were located in the dermis and surrounding the hair follicles of intact skin of *Itgbl1*$^{CreERT2/CreERT2}$::*ROSA26*$^{GRR/GRR}$ mice (Fig. 7F). To trace the migration of *Itgbl1*-expressing cells from intact skin during skin repair, we made wounds in these mice on Day 7, after the last tamoxifen injection. On Day 7 post injury, *Itgbl1*-positive cells had migrated to the wound margin but were not yet recruited to the center of the wound (Fig. 7G). On Day 14 post injury, *Itgbl1*-positive cells were observed diffusely arranged in the upper layer of the wound (Fig. 7H). These results indicated that a proportion of *Itgbl1*-expressing cells in intact skin may contribute to granulation tissue formation and scarring.

The data presented above provide experimental evidence that a proportion of myofibroblasts express *Itgbl1*, and that these cells might play important roles in skin wound healing. To further explore the

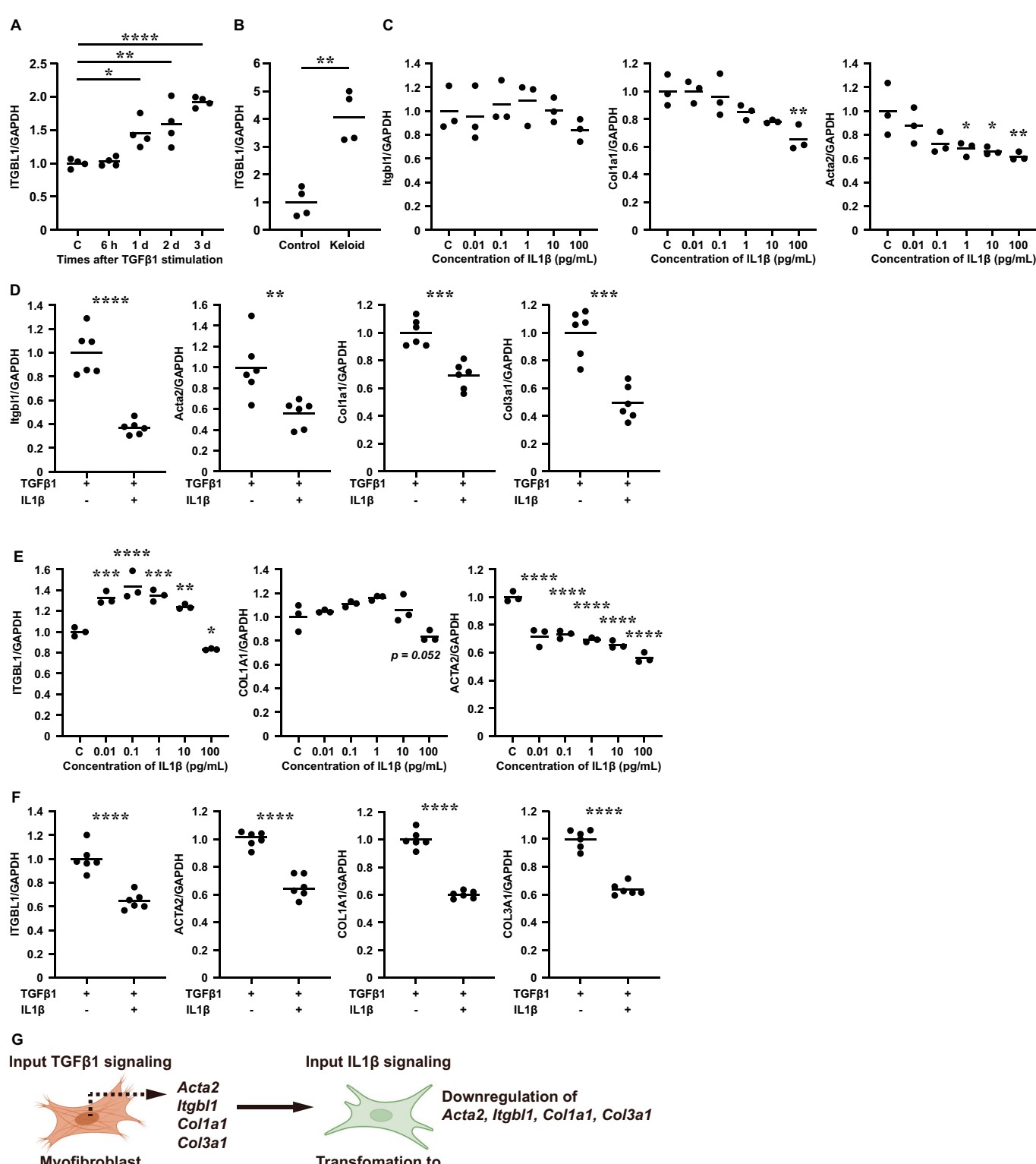

function of myofibroblast-derived *Itgbl1* in vivo, we generated mice with an *Itgbl1*-flox knock-in (*Itgbl1^flox/flox^*) using CRISPR-Cas technology as previously described (Tanimoto et al, 2022). These mice were crossed with *Tagln* Cre-reporter mice (*Tagln^Cre/Cre^*) to generate myofibroblast-specific *Itgbl1*-deficient mice (Fig. 7I). *Itgbl1* mRNA

expression at skin wound sites of *Itgbl1^flox/flox^::Tagln^Cre/Cre^* mice (0.058 ± 0.036) on Day 7 post injury was significantly decreased compared to *Itgbl1^flox/flox^* mice (1.0 ± 0.44; *P* < 0.0001) (Fig. 7J). The gross appearance of aseptic healing on Day 7 post injury showed a significant delay in *Itgbl1^flox/flox^::Tagln^Cre/Cre^* mice (64.4% ± 23) compared

**Figure 6.  Regulation of *Itgbl1* expression, myofibroblast differentiation, and fibrogenesis through antagonism between TGFβ1 and IL1β signaling.**

(A) qPCR measurement of the temporal expression of *ITGBL1* in HDF stimulated with TGFβ1 (100 pg/mL) ($n = 4$ independent cell cultures). *P* values: control (C) vs. 1 day (d) = 0.0121, C vs. 2 d = 0.0015, C vs. 3 d < 0.0001. (B) qPCR measurement of expression of *ITGBL1* in HDF normal and keloid-derived fibroblasts ($n = 4$ independent cell cultures). *P* value: Control vs. Keloid = 0.0011. (C) qPCR of *Itgbl1, Col1a1,* and *Acta2* expression in MDF stimulated with IL1β for 1 day ($n = 3$ independent cell cultures). *P* value (Col1a1/GAPDH): C vs. 100 pg/mL = 0.0032. *P* values (Acta2/GAPDH): C vs. 1 pg/mL = 0.0270, C vs. 10 pg/mL = 0.0186, C vs. 100 pg/mL = 0.0082. (D) qPCR of *Itgbl1, Acta2, Col1a1,* and *Col3a1* expression in MDF stimulated with TGFβ1 (100 pg/mL) and IL1β (100 pg/mL) for 1 day ($n = 6$ independent cell cultures). *P* values: Itgbl1/GAPDH <0.0001, Itgbl1/GAPDH = 0.0064, Col1a1/GAPDH = 0.0002, Col3a1/GAPDH = 0.0001. (E) qPCR of *ITGBL1, COL1A1,* and *ACTA2* expression in HDF stimulated with IL1β for 1 day ($n = 3$ independent cell cultures). *P* values (ITGBL1/GAPDH): C vs. 0.01 pg/mL = 0.0003, C vs. 0.1 pg/mL <0.0001, C vs. 1 pg/mL = 0.0001, C vs. 10 pg/mL = 0.0036, C vs. 100 pg/mL = 0.0363. *P* values (ACTA2/GAPDH): C vs. 0.01–100 pg/mL: <0.0001. (F) qPCR analysis of *ITGBL1, ACTA2, COL1A1,* and *COL3A1* expression in HDF stimulated with TGFβ1 (100 pg/mL) and IL1β (100 pg/mL) for 1 day ($n = 6$ independent cell cultures). *P* values: ITGBL1/GAPDH <0.0001, ACTA2/GAPDH <0.0001, COL1A1/GAPDH <0.0001, COL3A1/GAPDH <0.0001. (G) A model summarizing the interplay of signaling factors within dermal fibroblasts. Data information: All values represent the mean. Unpaired *t*-test (B, D, F) and one-way ANOVA followed by Dunnett's multiple comparisons test (control vs. sample) (A, C, E) were used to generate the indicated *P* values; *$P < 0.05$, **$P < 0.01$, ***$P < 0.001$, ****$P < 0.0001$. C control. Source data are available online for this figure.

with that in *Itgbl1^{flox/flox}* mice (41.7% ± 9.5; $P < 0.0001$) (Fig. 7K,L). This delayed healing was accompanied by increased granulation tissue area (Fig. 7M,N), suggesting phenotypic similarity to *Itgbl1^{−/−}* mice (Appendix Fig. S6B,C). However, there were no significant differences in collagen fibril morphology within intact skin from *Itgbl1^{flox/flox}* (44.26 ± 6.79 nm) and *Itgbl1^{flox/flox}::Tagln^{Cre/Cre}* (45.27 ± 9.47 nm) mice (Appendix Fig. S6D). Collectively, our findings indicate that: (1) expression of *Itgbl1* in *Acta2^{high}* myofibroblasts is important for granulation tissue formation; and (2) ITGBL1 is involved in regulating collagen assembly and fibrogenesis through myofibroblast differentiation-mediated antagonism of TGFβ1 and IL1β signaling from the proliferation to maturation phases of skin wound healing (Fig. 7O).

## Discussion

In this study, we integrated spatial transcriptome analysis with single-cell analysis to establish a novel, high-resolution spatial multiomics analysis method that reveals interactions among various cell types at different times and locations, while also identifying the major signal transduction pathways that mediate these interactions. Notably, our results demonstrate that a minority of time-specific *Acta2^{high}* fibroblasts play an important role in the fate of skin wound healing. Furthermore, by comparing wounds from WT and *PU.1^{−/−}* mice, we identified the inflammation-related scarring gene *Itgbl1* and showed that *Itgbl1* in *Acta2^{high}* myofibroblasts is important for granulation tissue formation.

In previous reports, the application of scRNA-Seq analysis has revealed the heterogeneity of dermal and wound fibroblasts in mice (Guerrero-Juarez et al, 2019; Mascharak et al, 2022) and humans (Ascensión et al, 2021; Theocharidis et al, 2022), while Foster and colleagues used integrated spatial multiomics to uncover the fate of fibroblasts (Foster et al, 2021). Thus far, no reports have utilized integrated spatial multiomics to analyze time-dependent changes in the comprehensive array of cell types and gene expression patterns involved in murine skin wound healing. Our development of the integrated spatial multiomics analysis method described in this paper has not yet reached the single-cell level of resolution, and so we are working on resolving technical challenges to advance this.

We have shown that *Acta2^{high}* myofibroblasts are critically important in the wound-healing process and in fibrosis development. *Acta2^{high}*-expressing myofibroblasts can cause pathologies such as

keloids (Deng et al, 2021). In the present study, we used single-cell analysis to demonstrate that *Acta2^{high}* myofibroblasts, which originate during the healing process, comprise a minority population of cells within wound granulation tissue. Additionally, using CellChat, we showed that inflammatory cell-derived IL1β might regulate fibroblast activation in the maturation phase. IL1β is a well-known pro-inflammatory cytokine that activates inflammatory cells and is upregulated in keloids (Ogawa, 2017). Wound healing in *Il1r1^{−/−}* mice, or those treated with an IL1 receptor antagonist, have previously shown reduced scarring (Thomay et al, 2009). Although many studies have been conducted regarding the activation of inflammation by IL1β, the molecular mechanisms of IL1β signaling in fibroblasts during tissue repair are largely unknown. Our study demonstrates that IL1β attenuates *Itgbl1, Col1α1, Col31α1,* and *Acta2* expression in TGF1β-stimulated fibroblasts, suggesting that IL1β participates in fibroblast transformation while antagonizing TGF1β signaling and inducing ECM protein expression during the granulation tissue formation and scarring processes.

Previously, in a comparison of wound tissue from WT versus *PU.1^{−/−}* mice, we identified inflammation-related mRNAs and miRNAs; we showed that one of these, *Spp1*, plays a critical role in scarring and can be molecularly targeted using therapeutic antisense oligodeoxynucleotides (Mori et al, 2008). In the present study, we identified *Spp1* as a positive control and revealed several gene candidates potentially involved in inflammation-related scarring. We identified the ECM gene *Itgbl1*, which is expressed in transforming *Acta2^{high}* and *Acta2^{low}* myofibroblasts affected by TGFβ1 and IL1β, and contributes to the regulation of collagen fibrillogenesis. Reportedly, *Spp1^{−/−}* mice exhibit less collagen organization into fibers than WT mice, resulting in smaller-diameter fibrils in their dermis (Liaw et al, 1998). The deposition of several ECM molecules, including collagen types I and III, depends on the presence and stability of fibronectin (Sottile et al, 2007). These observations indicate that fibronectin presence may affect the synthesis and/or turnover of other matrix components involved in the regulation of collagen fibril formation and bundling, either by affecting the types of collagen synthesized, or by associating with proteins present in the remodeling wound. In a recent study, the COL1A1 content at murine skin wound sites transfected with an *ITGBL1* overexpression plasmid was significantly increased after activation of dermal fibroblasts (Zou et al, 2022). Accordingly, we propose that collagen-binding ECM proteins regulate collagen fibrogenesis and may represent molecular targets for fibrosis therapy.

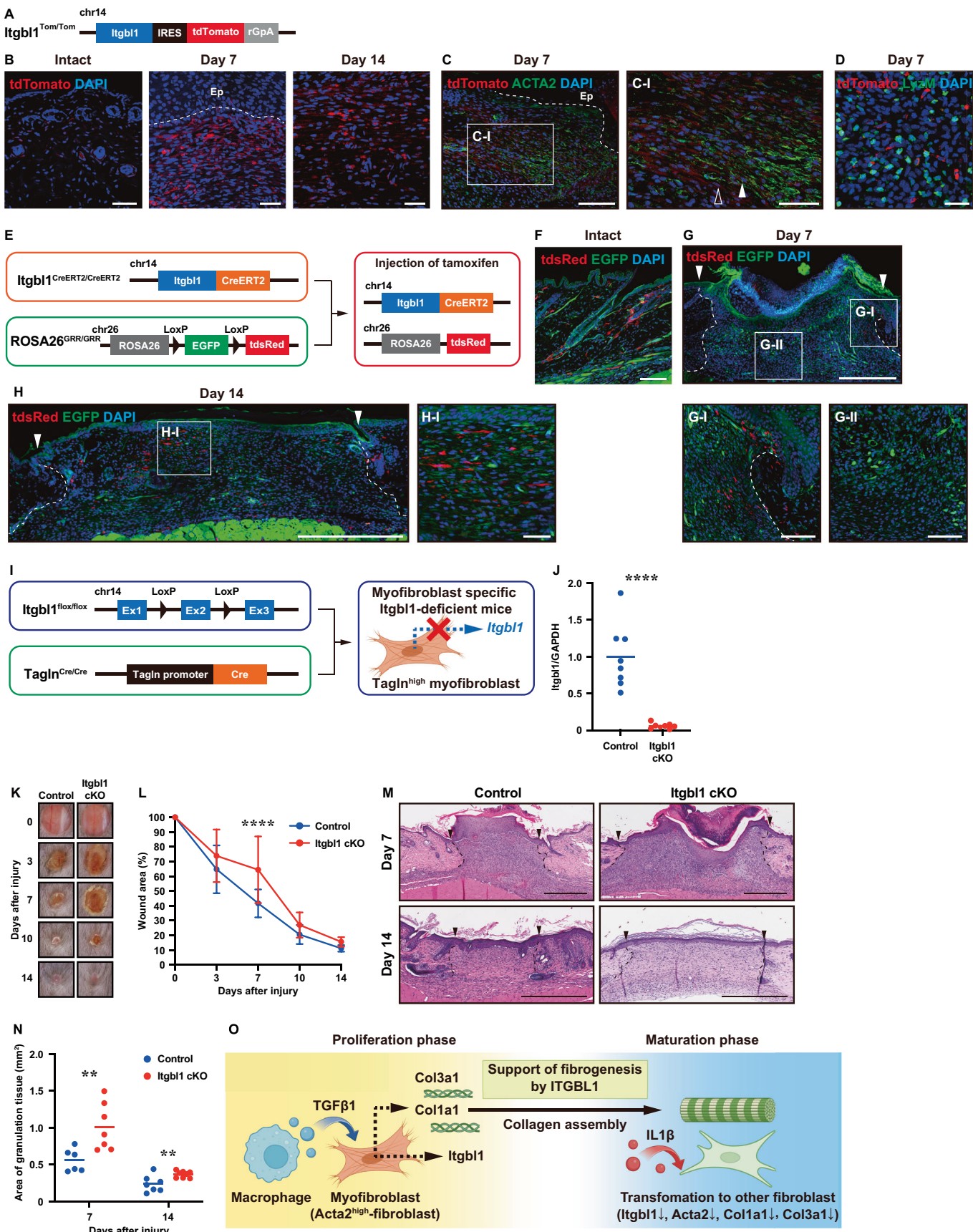

**Figure 7. Indispensable nature of *Itgbl1* in the formation of granulation tissue and support of fibrogenesis through myofibroblast differentiation-mediated antagonism of TGFβ1 and IL1β signaling.**

(A) Schematic diagram of the genomic construction of *Itgbl1*$^{Tom/Tom}$ mice. (B) Localization of *Itgbl1*-positive cells (red) at intact skin and wound sites in *Itgbl1*$^{Tom/Tom}$ mice. Ep epithelial cells. Scale bars, 50 μm. (C) IHC of ACTA2 at the skin wound site on Day 7 post injury in *Itgbl1*$^{Tom/Tom}$ mice. The white and empty arrowheads indicate ACTA2 and *Itgbl1* double-positive cells and *Itgbl1* single-positive cells, respectively. Scale bars; (C) 100 μm, (C–I) 50 μm. (D) Localization of macrophages (green) and *Itgbl1*-positive cells (red) at the skin wound site of *Itgbl1*$^{Tom/Tom}$::*LysM*$^{EGFP/EGFP}$ mice on Day 7 post injury. Scale bar, 50 μm. (E) Schematic diagram of the genomic construction of *Itgbl1*$^{CreERT2/}$ $^{CreERT2}$::*ROSA26*$^{GRR/GRR}$ mice. (F–H). Localization of *Itgbl1*-positive cells (red) on intact skin (F), and at Days 7 (G) and 14 (H) post injury. Arrowheads indicate the wound margin. Scale bars: (H) 500 μm; (F) 100 μm; (G-I, G-II, H-I) 50 μm. (I) Schematic diagram of the genomic construction of *Itgbl1*$^{flox/flox}$::*Tagln*$^{Cre/Cre}$ mice. (J) qPCR analysis of *Itgbl1* cKO in wound sites on Day 7 post injury confirming that *Itgbl1* cKO mice were significantly decreased in an expression of *Itgbl1* ($n = 8$). *P* value: Control vs. Itgbl1 cKO <0.0001. (K) Representative photographic images of the gross appearance of excisional wounds in control (*Itgbl1*$^{flox/flox}$) and *Itgbl1* conditional knockout (cKO, *Itgbl1*$^{flox/}$ $^{flox}$::*Tagln*$^{Cre/Cre}$) mice. (L) The proportion of the wound area remaining open at each time point relative to the initial wound area (Control, $n = 8$ wounds; *Itgbl1* cKO, $n = 16$ wounds). *P* value: Day 0 vs. Day 7 <0.0001. (M) H&E staining of wound site on Days 7 and 14 post injury (wound margin [arrowheads]). Scale bars: 500 μm. (N) Measurement of granulation tissue area on Days 7 and 14 post injury in control (Day 7; $n = 6$, Day 14; $n = 7$) and *Itgbl1* cKO mice (Day 7; $n = 7$, Day 14; $n = 8$). *P* values: Day 7 = 0.0094, Day 14 = 0.0100. (O) IL1β signaling, from the proliferation phase to the maturation phase, is involved in the transformation of activated myofibroblasts and contributes to normal mature scar formation. Data information: All values represent the mean ± SD (L). Two-way ANOVA followed by Sidak's multiple comparisons test (L) and unpaired *t*-tests (J, N) were used to generate the indicated *P* values; **$P < 0.01$, ****$P < 0.0001$.

Fibroblasts have long been recognized as the primary source of ECM. Nevertheless, recent studies have also shed light on the significant contributions of macrophages to ECM production (Chang et al, 2012; Schnoor et al, 2008) and scarring (Simões et al, 2020). During the transition period between the inflammatory and proliferation phases, macrophage-derived ECM may contribute to anchorage-dependent infiltration of fibroblasts to wound sites. Interestingly, our CellChat analysis revealed that some macrophage subclusters (M1, M3, M4, M6) expressed FN1, which aids in anchoring cells to the dermal ECM and is involved in cell adhesion and migration (Patten and Wang, 2021). These subclusters also interact with inflammatory cells and minority fibroblast subclusters on Day 3 post injury. These findings suggest that macrophages, acting in the early stages after injury, promote the initiation of granulation tissue formation to support migration, proliferation, and transformation of fibroblasts from intact skin. We are further investigating this possible role for macrophages.

Looking ahead, we aim to further refine these multiomics methods to achieve even higher resolution of gene expression in single cells within wound tissue. This refinement will ultimately illuminate the intricate relationships between inflammation, tissue repair, and the genes involved in fibrosis. We anticipate that the findings of this study will contribute to a deeper understanding of these biological processes and drive the development of clinical applications.

# Methods

### Reagents and tools table

| Reagent/resource | Reference or source | Identifier or catalog number |
|---|---|---|
| **Experimental models** | | |
| C57BL/6J (*M. musculus*) | Jackson Lab Japan | 000664 |
| Crl:CD1(ICR) (*M. musculus*) | Jackson Lab Japan | 022 |
| PU.1$^{-/-}$, C57BL/6J (*M. musculus*) | de Kerckhove et al, 2018 | N/A |
| Itgbl1$^{-/-}$, C57BL/6J (*M. musculus*) | This study | C57BL/6-Itgbl1$^{em4(null)Rmori}$ |

| Reagent/resource | Reference or source | Identifier or catalog number |
|---|---|---|
| Itgbl1$^{(flox)}$, C57BL/6J (*M. musculus*) | This study | C57BL/6-Itgbl1$^{em2(flox)Rmori}$ |
| Itgbl1$^{(tdTomato)}$, C57BL/6J (*M. musculus*) | This study | C57BL/6-Itgbl1$^{em1(tdTomato)Rmori}$ |
| Itgbl1$^{(Cre/ERT2)}$, C57BL/6J (*M. musculus*) | This study | C57BL/6-Itgbl1$^{em3(Cre/ERT2)Rmori}$ |
| LysM$^{(EGFP)}$, C57BL/6J (*M. musculus*) | Faust et al, (2000) | Prof. Thomas Graf, CRG, Spain |
| ROSA26$^{(GRR)}$, C57BL/6N (*M. musculus*) | RIKEN BRC | RBRC04874 |
| Tagln$^{(Cre)}$, C57BL/6J (*M. musculus*) | Jackson Lab | 7491 |
| Murine primary dermal fibroblasts (*M. musculus*) | This study | N/A |
| Human primary dermal fibroblasts (*H. sapiens*) | Thermo Fisher Scientific | C0135C |
| Human skin-derived fibroblast (*H. sapiens*) | This study | N/A |
| Keloid-derived fibroblasts (*H. sapiens*) | This study | N/A |
| **Recombinant DNA** | | |
| pIRES-creERT2-Itgbl1 | This study | N/A |
| pIRES-tdTomato-Itgbl1 | This study | N/A |
| pflox-Itgbl1 | This study | N/A |
| **Antibodies** | | |
| alpha smooth muscle Actin | abcom | ab5694 |
| Goat anti-rabbit-Alexa488 | Thermo Fisher Scientific | A-11008 |
| **Oligonucleotides and other sequence-based reagents** | | |
| TaqMan Probe, Itgbl1 (*M. musculus*) | Applied Biosystems | Assay ID: Mm00520935_m1 |
| TaqMan Probe, ITGBL1 (*H. sapiens*) | Applied Biosystems | Assay ID: Hs01557019_m1 |
| TaqMan Probe, Acta2 (*M. musculus*) | Applied Biosystems | Assay ID: Mm00725412_s1 |
| TaqMan Probe, ACTA2 (*H. sapiens*) | Applied Biosystems | Assay ID: Hs00426835_g1 |

| Reagent/resource | Reference or source | Identifier or catalog number |
|---|---|---|
| TaqMan Probe, Col1a1 (*M. musculus*) | Applied Biosystems | Assay ID: Mm00801666_g1 |
| TaqMan Probe, COL1A1 (*H. sapiens*) | Applied Biosystems | Assay ID: Hs00164004_m1 |
| TaqMan Probe, Col3a1 (*M. musculus*) | Applied Biosystems | Assay ID: Mm00802300_m1 |
| TaqMan Probe, COL3A1 (*H. sapiens*) | Applied Biosystems | Assay ID: Hs00943809_m1 |
| TaqMan Probe, Gapdh (*M. musculus*) | Applied Biosystems | Assay ID: Mm99999915_g1 |
| TaqMan Probe, GAPDH (*H. sapiens*) | Applied Biosystems | Assay ID: Hs02786624_g1 |
| PCR primer | This study | Dataset EV11 |
| Alt-R CRISPR-Cas9 crRNA | IDT | Dataset EV11 |
| Alt-R CRISPR-Cas9 tracrRNA | IDT | 1072533 |
| **Chemicals, enzymes, and other reagents** | | |
| KOD-FX | TOYOBO | KFX-101 |
| Tamoxifen | Merck | T5648-1G |
| Corn oil | Sigma-Aldrich | C8267-500ML |
| Whole skin dissociation kit | Miltenyi Biotec | 130-101-540 |
| Lysis Buffer | BD Biosciences | 555899 |
| Debris Removal Solution | Miltenyi Biotec | 130-109-398 |
| LIVE/DEAD Viability/Cytotoxicity Kit for mammalian cells | Thermo Fisher Scientific | L3224 |
| Countess Cell Counting Chamber Slides | Thermo Fisher Scientific | C10228 |
| Chromium Next GEM Single Cell 3' Reagent Kit v3.1 Dual Index | 10× Genomics | 1000269 |
| Visium Spatial Gene Expression Slides & Reagents Kit | 10× Genomics | 1000187 |
| Visium Accessory Kit | 10× Genomics | 1000194 |
| Dual Index Kit TT Set A | 10× Genomics | 1000215 |
| Tissue-Tek OCT Compound | Sakura Finetek Japan | 4583 |
| Water-deionized and sterilized | Nacalai Tesque | 06442-95 |
| Ethanol (99.5) nuclease and protease tested | Nacalai Tesque | 08948-25 |
| Isopentane | Nacalai Tesque | 26404-75 |
| 4% paraformaldehyde solution | Nacalai Tesque | 09154-85 |
| Picro-Sirius Red Stain Kit (For Collagen) | ScyTek Laboratories | PSR-1 |
| RNAscope 2.5 HD Reagent Kit-RED | Advanced Cell Diagnostics | 322350 |
| miRNeasy Tissue/Cells Advanced Mini Kit | Qiagen | 217684 |
| TruSeq stranded mRNA sample prep kit | Illumina | 20020595 |

| Reagent/resource | Reference or source | Identifier or catalog number |
|---|---|---|
| T-PER Tissue Protein Extraction Reagent | Thermo Fisher Scientific | 78510 |
| Protease inhibitor | Nacalai Tesque | 25955-24 |
| IL1β mouse ELISA kit | R&D systems | MLB00C |
| QuickZyme Hydroxyproline Assay kit | QuickZyme Bioscience | QZBHYPRO1 |
| 2-mL polypropylene-lined screw cap microtubes | Sarstedt | 72694-100002 |
| 6 mol/L Hydrochloric acid | FUJIFILM Wako Chemicals | 084-05425 |
| Human Fibroblast Expansion Basal Medium | Thermo Fisher Scientific | M106500 |
| Low Serum Growth Supplement | Thermo Fisher Scientific | S-003-10 |
| Dispase I | FUJIFILM Wako Chemicals | 386-02271 |
| Recombinant Mouse TGFβ1 protein | R&D Systems, | 7666-MB |
| HumanKine recombinant human TGFβ1 protein | Proteintech Group | HZ-1011 |
| HumanKine recombinant human IL1β protein | Proteintech Group | HZ-1164 |
| **Software** | | |
| BioTuring Browser | BioTuring | N/A |
| NIS-Elements C software version 4.13 | Nikon Solutions | N/A |
| AR software version 4.0 | Nikon Solutions | N/A |
| IMARIS 7.7.2 | Oxford Instruments | N/A |
| ImageJ | https://imagej.net/ij/ | N/A |
| Loupe Browser software | 10× Genomics | N/A |
| CASAVA 1.8.2 software | Illumina | N/A |
| Cuffnorm version 2.2.1 | Trapnell et al, 2010 | |
| GraphPad Prism software version 10 | GraphPad Software | N/A |
| **Other** | | |
| Charles River-LPF diet | Oriental Yeast | CR-LPF |
| Dermo Camera | CASIO COMPUTER | DZ-D100 |
| 4-mm biopsy punch | Kai Industries | BP-40F |
| 6-mm biopsy punch | Kai Industries | BP-60F |
| PI-200 | Kurabo Industries | N/A |
| gentleMACS Dissociator | Miltenyi Biotec | 130-093-235 |
| Countess II FL Automated Cell Counter | Thermo Fisher Scientific | AMQAF1000 |
| Chromium Controller | 10× Genomics | 1000202 |

| Reagent/resource | Reference or source | Identifier or catalog number |
|---|---|---|
| DNBSEQ-G400RS (MGISeq2000) DNBSEQ-G400RS is the original product name as MGISEQ-2000RS. | MGI Tech | N/A |
| Confocal microscopy C2+ | Nikon Solutions | N/A |
| TissueLyzer II | Qiagen | 85300 |
| QuantStudio 12 K Flex | Applied Biosystems | QS-OA03 |
| ThermoMax microplate reader | Molecular Devices | 8254-30-1012 |

## Mice and the wound model

All experiments were conducted in accordance with the provisions of the Ethics Review Committee for Animal Experimentation at Nagasaki University. All adult mice were kept in a barrier facility (temperature; 22–25 °C, 12 h light/dark cycle) under specific-pathogen-free conditions. Mice were fed ad libitum with the Charles River-LPF diet (360 kcal/100 g body weight; 13% fat, 26% protein, and 61% carbohydrate; Oriental Yeast, Tokyo, Japan). C57BL/6J age-matched male mice (age, 7–12 weeks) (Jackson Laboratory Japan Inc., Kanagawa, Japan) and other *Itgbl1* transgenic mice (age, 7–12 weeks) were anesthetized; four full-thickness excisional wounds were aseptically made in the shaved dorsal skin using a 4-mm biopsy punch (Kai Industries, Gifu, Japan). Wound tissues were harvested with a 6-mm biopsy punch.

The *PU.1*$^{-/-}$ mouse line was described previously (de Kerckhove et al, 2018). One-day-old pups received anesthesia and full-thickness 1-cm incisional wounds in the dorsal skin. The wounds were harvested 24 h after injury.

C57BL/6-*Itgbl1*$^{em2(flox)Rmori}$ mice were crossed with Transgelin (*Tagln*)-Cre knock-in mice (stock no. 17491; Jackson Laboratory, ME, USA) to generate conditional knockout mice with *Tagln*-expressing cells.

C57BL/6-*Itgbl1*$^{em1(tdTomato)Rmori}$ mice were crossed with *LysM-EGFP* mice (de Kerckhove et al, 2018) to generate *Itgbl1*$^{Tom/Tom}$::*LysM*$^{EGFP/EGFP}$ mice.

C57BL/6-*Itgbl1*$^{em3(Cre/ERT2)Rmori}$ mice were crossed with green-to-red fluorescent-convertible Cre-reporter mice (*ROSA26*$^{GRR/GRR}$, stock no. RBRC04874; RIKEN BRC, Ibaraki, Japan) (Hasegawa et al, 2013) to generate *Itgbl1*$^{CreERT2/CreERT2}$::*ROSA26*$^{GRR/GRR}$ mice.

For wound area analysis, each wound region was digitally photographed without blinding using a Dermo Camera (CASIO COMPUTER Co. Ltd., Tokyo, Japan) at the indicated time intervals. Digital images of the wound areas were measured using ImageJ software.

## Generation of whole-body *Itgbl1*$^{-/-}$ and *Itgbl1*$^{flox/+}$ mice

Two mouse genomic sequences (5′-GTCTGATCCCA-TAGCCTGTA-3′ and 5′-CACAACTAGAGTTACCTTGA-3′) from introns 1 and 2 of *Itgbl1* were selected as the CRISPR target. We purchased synthetic CRISPR RNA (crRNA) containing this target sequence from Integrated DNA Technologies Inc. (Coralville, IA, USA). The flox donor plasmid, *pflox-Itgbl1*, carried the genomic region from 207 base pairs (bp) upstream of exon 1 to 1246 bp

downstream of exon 2 of *Itgbl1*. Two loxp sequences were inserted in the donor vector, one at 548 bp upstream and one at 343 bp downstream of exon 2 of the gene.

The CRISPR-Cas9 ribonucleoprotein complex and each donor DNA were microinjected into zygotes of C57BL/6J mice (Jackson Laboratory Japan, Kanagawa, Japan), in accordance with our previous report (Tanimoto et al, 2022). Subsequently, microinjected zygotes were transferred into the oviducts of pseudopregnant female ICR mice (Jackson Laboratory Japan), and newborns were obtained.

To confirm the designed flox allele, genomic DNA was purified from the tail using PI-200 (Kurabo Industries Ltd., Osaka, Japan), in accordance with the manufacturer's protocol. Genomic PCR was performed with KOD-Fx (Toyobo, Osaka, Japan). Primers *Itgbl1* dflox long inLeF (5′-ACAGGAAAGAATAGCGTGGATTTGGAA C-3′) and *Itgbl1* dflox long RiR (5′-ACCTATGATTTGTTGCATG CTCAGGAAG-3′) were used to detect the knockout and flox alleles. Additionally, we checked the random integration of donor DNA by PCR amplification of the ampicillin resistance gene using primers Amp detection-F (5′-TTGCCGGGAAGCTAGAGTAA-3′) and Amp detection-R (5′-TTTGCCTTCCTGTTTTTG CT-3′).

## Generation of knock-in *Itgbl1*-tdTomato mice and *Itgbl1*-CreERT2 knock-in mice

We selected a sequence (5′-AATATCCTTAATGATCCCAT-3′) encompassing the termination codon of *Itgbl1* as the CRISPR target and purchased synthetic crRNA containing this target from Integrated DNA Technologies. In the *pItgbl1-tdTomato* donor DNA, we placed the IRES-tdTomato-rabbit beta-globin polyadenylation (*rGpA*) signal sequence between the 5′ and 3′ homology arms. We constructed *pItgbl1-CreERT2* donor DNA using the same approach: the 5′-homology arm is the genomic region from 1021 bp upstream to 6 bp downstream of the *Itgbl1* termination codon, and the 3′-homology arm is the genomic region from 7 bp downstream to 925 bp downstream of the *Itgbl1* termination codon.

The CRISPR-Cas9 ribonucleoprotein complex and each donor DNA were microinjected into zygotes of C57BL/6J mice (Jackson Laboratory Japan), in accordance with our previous report (Tanimoto et al, 2022). Subsequently, microinjected zygotes were transferred into the oviducts of pseudopregnant female ICR mice (Jackson Laboratory Japan), and newborns were obtained.

To confirm the designed knock-in allele, genomic DNA was purified from the tail using PI-200 (KURABO INDUSTREIS LTD., Osaka, Japan), in accordance with the manufacturer's protocol. Genomic PCR was performed with KOD-Fx (Toyobo). Primers *Itgbl1* screening 5F (5′-AGCTCCCCTCTGAGAAGCAGCTATTC AA-3′) and *Itgbl1* screening 3Rv (5′-TGCAAGAGAACCCCAAAT ATTGTACATT-3′) were used to detect the correct knock-in allele. Additionally, we checked the random integration of donor DNA by PCR amplification of the ampicillin resistance gene using primers Amp detection-F (5′-TTGCCGGGAAGCTAGAGTAA-3′) and Amp detection-R (5′-TTTGCCTTCCTGTTTTTGCT-3′).

## Tamoxifen treatment

Tamoxifen (Merck KGaA, Darmstadt, Germany) was prepared as a 20 mg/mL stock solution in corn oil. The administration of corn oil

alone was used as vehicle control. Mice were intraperitoneally administered tamoxifen (~30 g/body weight) from a 2 mg/100 µL solution for 4 consecutive days. Tissues were harvested on Day 7, after the last injection of tamoxifen. Tamoxifen administration did not alter the body weight or condition of the mice.

## Single-cell analysis

Harvested wound tissue (4-mm diameter) was pooled from four samples and dissociated using a whole skin dissociation kit (Miltenyi Biotec Inc., Bergisch Gladbach, Germany), in accordance with the manufacturer's instructions. Briefly, tissues were minced and incubated in enzymes for 1 h at 37 °C. After incubation, cell aggregates were mechanically dissociated using a gentleMACS Dissociator (Miltenyi Biotec Inc.). Single-cell suspensions were treated with Lysis Buffer (BD Biosciences). Dead cells and debris were removed using Debris Removal Solution (Miltenyi Biotec Inc.). Live cells were detected with a LIVE/DEAD Viability/ Cytotoxicity Kit for mammalian cells (Thermo Fisher Scientific, MA, USA) and a Countess II FL Automated Cell Counter (Thermo Fisher Scientific).

Single-cell suspensions and unsorted cells were captured by a droplet-based device (Chromium Controller, 10× Genomics, Pleasanton, CA, USA). The cDNA libraries were constructed using a Chromium Next GEM Single Cell 3' Reagent Kit v3.1 (Dual Index; 10× Genomics). Sequencing was performed on a DNBSEQ-G400RS platform (MGI Tech Co. Ltd., Shenzhen, China). BioTuring Browser (BioTuring Inc., San Diego, CA, USA) was used to perform downstream functional analysis of scRNA-Seq data.

## Spatial transcriptome analysis

Comprehensive spatial transcriptome analysis was performed using Visium Spatial Gene Expression Solution (10× Genomics), in accordance with the manufacturer's instructions. Briefly, harvested wound tissue (Day 3, Day 7, and Day 14; $n = 1$) was fixed by isopentane, placed in liquid nitrogen, and embedded with Tissue-Tek OCT Compound (Sakura Finetek Japan Co. Ltd., Tokyo, Japan). Each frozen slice (10-µm thickness) was placed on a Tissue Optimization Slide and a Gene Expression Slide containing the spatial barcode, unique molecular identifier, and poly deoxythymidine (dT) arrayed on each 50-µm diameter spot. Slices were stained with hematoxylin and eosin (H&E) and were permeabilized to captured mRNAs by poly(dT), after which the cDNA library was constructed for RNA sequencing (RNA-Seq). RNA-Seq was performed on the MGISeq2000 platform in 28 + 100 or 28 + 120 base paired-end mode. Raw sequencing data were converted to expression matrices using Space Ranger (version spaceranger-1.3.0, 10× Genomics). H&E-stained image data and RNA-Seq data were overlapped to visualize mRNA expression while maintaining tissue morphology. Finally, data were analyzed with an automatic clustering algorithm of the Loupe Browser software (10× Genomics), and pathologic annotations were made directly on H&E images.

## Histology

Harvested tissues were fixed in 4% paraformaldehyde (PFA) overnight and embedded in paraffin or Tissue-Tek OCT

Compound. All specimens were cut into 6-µm-thick sections and subjected to H&E staining, in situ hybridization (ISH), picrosirius red staining, and immunohistochemistry for myofibroblast using anti-alpha smooth muscle antibody (Abcom, ab5694). Areas of granulation tissue was quantified as previously described (de Kerckhove et al, 2018). Observations were made via confocal microscopy on a C2+ system (Nikon Solutions, Tokyo, Japan). NIS-Elements C software version 4.13 (Nikon Solutions), AR software version 4.0 (Nikon Solutions), or IMARIS 7.7.2 (Oxford Instruments, Oxford, UK) were used for data analysis.

## ISH

ISH was performed using RNAscope (Advanced Cell Diagnostics Inc., Hayward CA, USA), in accordance with the manufacturer's instructions. Briefly, 4% PFA perfusion-fixed tissues were embedded in paraffin. Sections (6-µm) were deparaffinized and incubated with hydrogen peroxide for 10 min at room temperature. After washing with distilled water, sections were treated with target retrieval solution for 15 min at 100 °C (mild boiling) and re-washed with distilled water. Sections were incubated with 100% EtOH for 5 min, dried for 15 min at 60 °C, treated with protease for 30 min at 40 °C, and washed in distilled water. Sections were treated with in situ probes for 2 h at 40 °C and developed using Fast Red. Counter-staining was conducted using Gill's hematoxylin.

## TEM and morphological analysis of collagen

TEM and morphological analysis of collagen were performed as previously described (Mori et al, 2008).

## mRNA sequencing

Harvested tissues were homogenized using TissueLyzer II (Qiagen, Venlo, the Netherlands) and RNA was extracted/purified using a miRNeasy Tissue/Cells Advanced Mini Kit (Qiagen), in accordance with the manufacturer's instructions. Library preparation for mRNA sequencing was performed using a TruSeq stranded mRNA sample prep kit (Illumina), in accordance with the manufacturer's instructions. Sequencing was performed on a DNBSEQ-G400RS platform (MGI Tech Co. Ltd., Shenzhen, China) in a 75-base single-end mode. Illumina CASAVA 1.8.2 software was used for base calling. Sequenced reads were mapped to the mouse mm10 reference genome. The fragments per kilobase of exon per million mapped fragments (FPKMs) were calculated using Cuffnorm version 2.2.1.

## CellChat receptor-ligand analysis

Potential cell–cell interactions at skin wound sites were clarified using the CellChat platform (https://github.com/sqjin/CellChat) (Jin et al, 2021). A CellChat object was created with normalized scRNA-Seq data to infer cell communication between clusters (corresponding to subpopulations of macrophages, fibroblasts, dendritic cells, neutrophils, B cells, T cells, pericytes, epithelial cells, endothelial cells, mesenchymal cells, and Schwann cells) and to explore the differences between two samples (Days 7 and 14). CellChatDB mouse was used as a database of ligand-receptor interactions.

## Cell-type deconvolution and cell–cell communication analysis for spatial transcriptome data

To elucidate the cellular composition at skin wound sites measured by the Visium platform, cell-type deconvolution was performed using the RCTD algorithm implemented in the spacexr package (https://github.com/dmcable/spacexr). The scRNA-Seq data, previously annotated with 11 cell types (identical to CellChat analysis), served as the reference; the ratio of cell types on each Visium spot was calculated. Visualization of cellular composition was then performed using the "SpatialFeaturePlot" function in the Seurat package. Analyses of cell–cell communication for signal direction at skin wound sites were performed using the COMMOT platform (https://github.com/zcang/COMMOT) (Cang et al, 2023).

## qPCR

qPCR was performed as previously described (de Kerckhove et al, 2018). Briefly, harvested tissues were homogenized by TissueLyzer II (Qiagen), and total mRNA samples were extracted/purified using an miRNeasy Tissue/Cells Advanced Kit (Qiagen), according to the manufacturer's instructions. mRNA quantification was performed with TaqMan Gene Expression Assays (Applied Biosystems, Foster City, CA, USA) using a QuantStudio 12K Flex (Applied Biosystems).

## Enzyme-linked immunosorbent assays (ELISAs)

Total protein extraction was performed as previously described (de Kerckhove et al, 2018). Briefly, harvested tissues were homogenized using a TissueLyzer II (Qiagen) and T-PER Reagent (Thermo Fisher Scientific) containing proteinase inhibitors. Sample proteins were filtered using an Ultrafree-MC 0.45-µm filter (Millipore). IL1β concentrations were measured with an IL1β mouse ELISA kit (R&D Systems, Minneapolis, MN, USA).

## Hydroxyproline analysis

Hydroxyproline assays were performed using the QuickZyme Hydroxyproline Assay kit (QZBHYPRO1, QuickZyme Bioscience, Netherlands), in accordance with the manufacturer's instructions. In summary, samples were placed in 2-mL polypropylene-lined screw cap microtubes (#72694-100002, Sarstedt, Germany) and mixed with 200 µL of 6 M HCl per tube, then subjected to brief centrifugation. The tubes were then incubated at 95 °C for 20 h. After the tubes had been cooled to room temperature, they were centrifuged at $13{,}000 \times g$ for 10 min to separate the supernatant from precipitated particles. The resulting supernatants were diluted with 0.5 volume of distilled water for adjustment to 4 M HCl. For the assay, samples were further diluted at a 1:20 ratio with 4 M HCl. Then, 35 µL of the diluted samples and standards were pipetted into separate microplate wells. Each well received 75 µL of assay buffer; the plate was sealed and incubated at room temperature for 20 min with agitation to ensure complete mixing. Subsequently, 75 µL of detection reagents were added to each well, and the plate was incubated at 60 °C for 1 h; this was followed by cooling to room temperature. Absorbance at 570 nm was measured with a ThermoMax microplate reader (8254-30-1012, Molecular Devices, CA, USA).

## Cell culture

Murine primary dermal fibroblasts were isolated as follows. Briefly, shaved murine dorsal skin tissue ($2 \text{ cm}^2$) was incubated in Gibco Human Fibroblast Expansion Basal Medium supplemented with Gibco Low Serum Growth Supplement (Thermo Fisher Scientific) including dispase I (500 unit/mL; FUJIFILM Wako Chemicals, Tokyo, Japan) and Gibco Antibiotic-Antimycotic (Thermo Fisher Scientific) for 16 h at 4 °C. The epidermis was detached, and dermis tissue was minced to 1 mm². Minced tissue was dissociated using a Whole Skin Dissociation Kit, human (Miltenyi Biotech Inc.), in accordance with the manufacturer's instructions. Murine primary dermal fibroblasts were cultured using a Gibco Human Fibroblast Expansion Basal Medium supplemented with Gibco Low Serum Growth Supplement (Thermo Fisher Scientific).

Adult human primary dermal fibroblasts (C0135C; Thermo Fisher Scientific) were cultured using Gibco Human Fibroblast Expansion Basal Medium supplemented with Gibco Low Serum Growth Supplement (Thermo Fisher Scientific).

Murine and human primary fibroblasts were treated with Recombinant Mouse TGFβ1 protein (R&D Systems), HumanKine recombinant human TGFβ1 protein, and IL1β protein (Proteintech Group Inc., Rosemont, IL, USA).

## Human skin samples

Human skin samples were harvested from adult male and female Japanese patients during elective plastic surgery, and diagnosis of keloid was confirmed by routine pathological examination. All procedures involving human subjects were approved by the Keio University Institutional Review Board (IRB No. 1101-116-353), and volunteers provided written informed consent. This study was conducted in accordance with the World Medical Association Declaration of Helsinki and the Department of Health and Human Services Belmont Report.

## Statistical analysis

Data are expressed as means ± standard deviations. The statistical significance of differences between means was assessed by analysis of variance (ANOVA) followed by Tukey's test for multiple comparisons, Dunnett's multiple comparisons test, or an unpaired Student's $t$-test followed by Welch's test for analysis of two groups only. Multiple comparison tests were performed using two-way ANOVA followed by the Sidak multiple comparisons test in GraphPad Prism software (GraphPad Software, San Diego, CA, USA).

# Data availability

The reported Visium, scRNA-Seq, and mRNA-Seq data are available in the NCBI Gene Expression Omnibus under accession no. GSE234272 (https://www.ncbi.nlm.nih.gov/geo/query/acc.cgi?acc=GSE234272).

The source data of this paper are collected in the following database record: biostudies:S-SCDT-10_1038-S44319-024-00322-3.

## Peer review information

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

## Acknowledgements

We acknowledge the NGS core facility at the Research Institute for Microbial Diseases of Osaka University for the sequencing and Visium data analysis. This work was supported in part by the Japan Society for the Promotion of Science (Grant-in-Aid for Scientific Research [A], 21H04838 to RM; Grant-in-Aid for Scientific Research [B], 22H03251 to SEK; Grant-in-Aid for Scientific Research [B], 23H02980 to DO; Grant-in-Aid for Scientific Research [C], 22K12246 to SS), the Atomic Bomb Disease Institute Nagasaki University to RM and MN, and the Japan Agency for Medical Research and Development–Core Research for Evolutional Science and Technology (AMED–CREST) (22gm1810003h0001) to DO.

## Author contributions

**Sang-Eun Kim**: Investigation. **Ryota Noda**: Investigation. **Yu-Chen Liu**: Software. **Yukari Nakajima**: Investigation. **Shoichiro Kameoka**: Software. **Daisuke Motooka**: Software. **Seiya Mizuno**: Resources; Investigation. **Satoru Takahashi**: Resources; Investigation. **Kento Takaya**: Investigation. **Takehiko Murase**: Investigation. **Kazuya Ikematsu**: Investigation. **Katsiaryna Tratsiakova**: Investigation. **Takahiro Motoyama**: Investigation. **Masahiro Nakashima**: Investigation. **Kazuo Kishi**: Investigation. **Paul Martin**: Resources; Writing—review and editing. **Shigeto Seno**: Software; Funding acquisition. **Daisuke Okuzaki**: Resources; Software; Funding acquisition. **Ryoichi Mori**: Conceptualization; Data curation; Formal analysis; Supervision; Funding acquisition; Validation; Investigation; Visualization; Methodology; Writing—original draft; Project administration; Writing—review and editing.

Source data underlying figure panels in this paper may have individual authorship assigned. Where available, figure panel/source data authorship is listed in the following database record: biostudies:S-SCDT-10_1038-S44319-024-00322-3.

## Disclosure and competing interests statement

The authors declare no competing interests.

# Expanded View Figures

**Figure EV1. Identification of top ten marker genes in the macrophage subclusters.**

Bubble heatmap of macrophage subclusters (M1–M6) and their respective marker genes.

▶

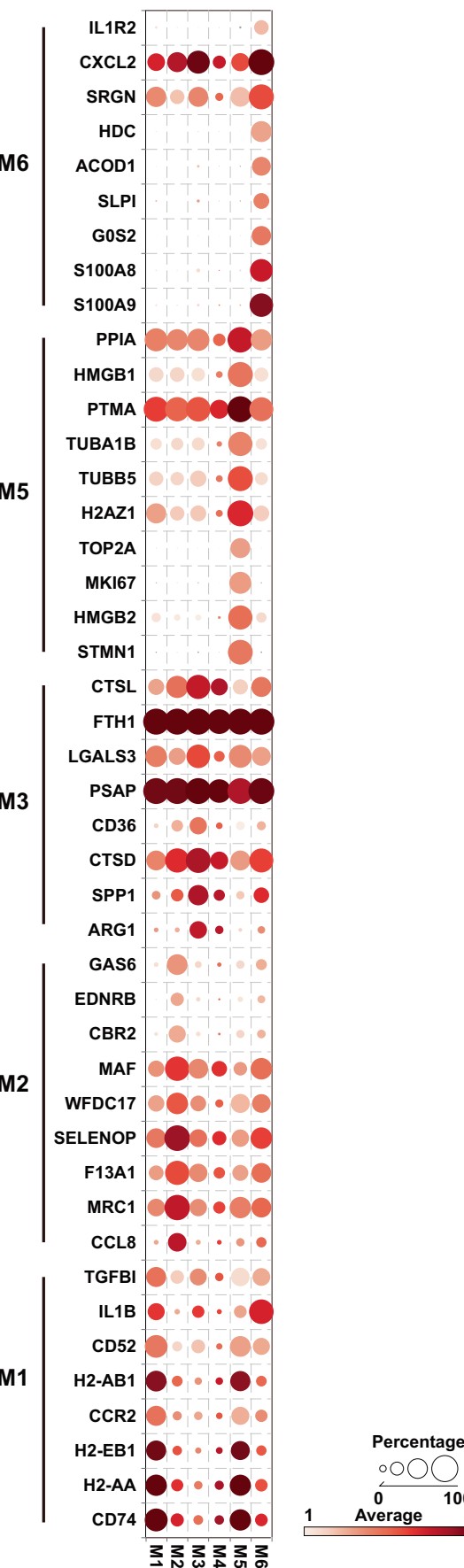

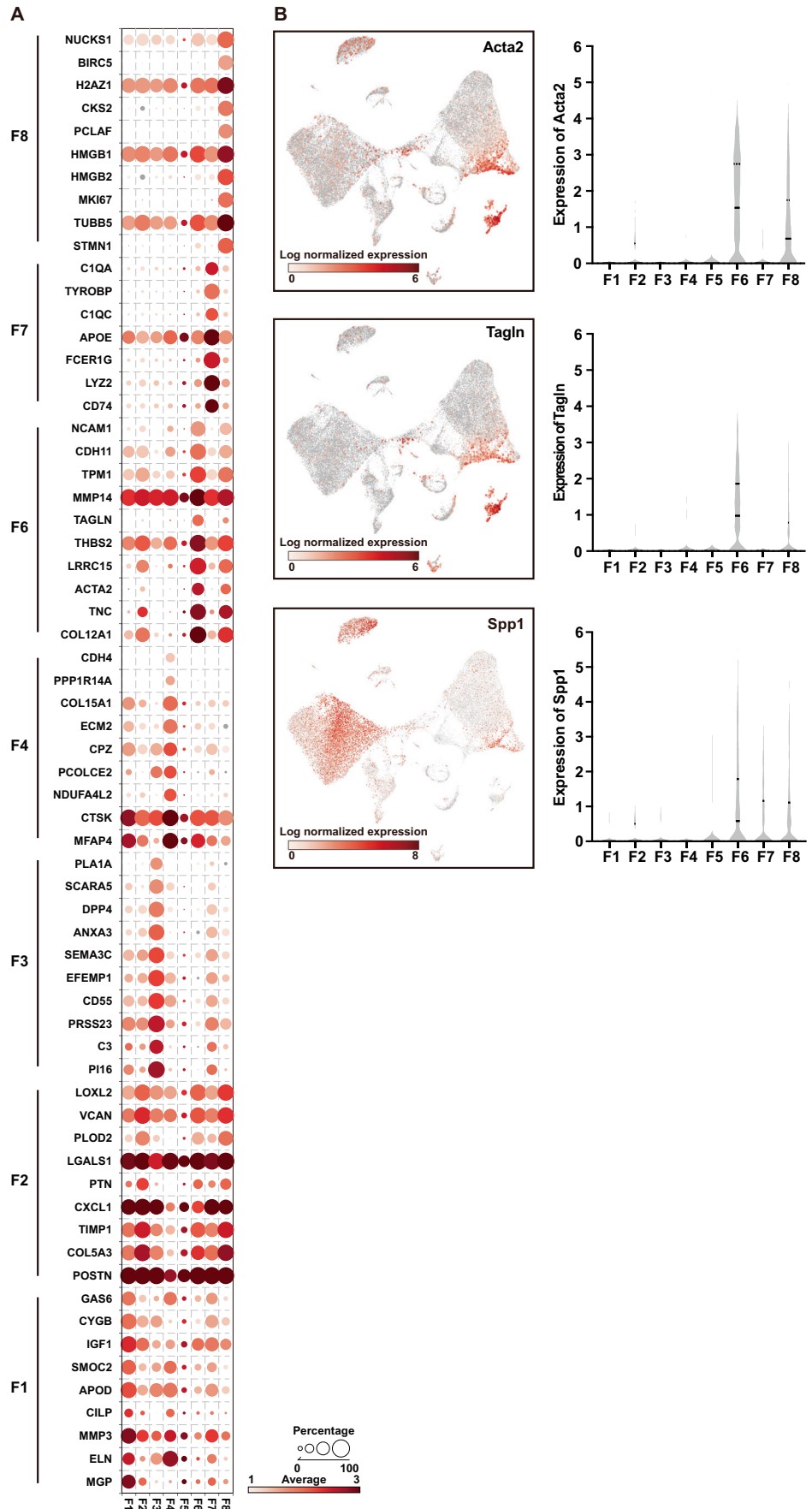

Figure EV2. Identification of top 10 marker genes in the fibroblast subclusters.

(A) Bubble heatmap of fibroblast subclusters (F1–F8) and their respective marker genes. (B) UMAP plot (left) of expression levels of *Acta2* (top), *Tagln* (middle), and *Spp1* (bottom) across 40,024 cells from Days 3, 7, and 14 post injury (left panel). Violin plot (right) showing expression levels of these genes in each fibroblast subcluster. F1: $n = 4265$; F2: $n = 3849$; F3: $n = 2218$; F4: $n = 1421$; F5: $n = 1369$; F6: $n = 977$; F7: $n = 804$; F8: $n = 532$.

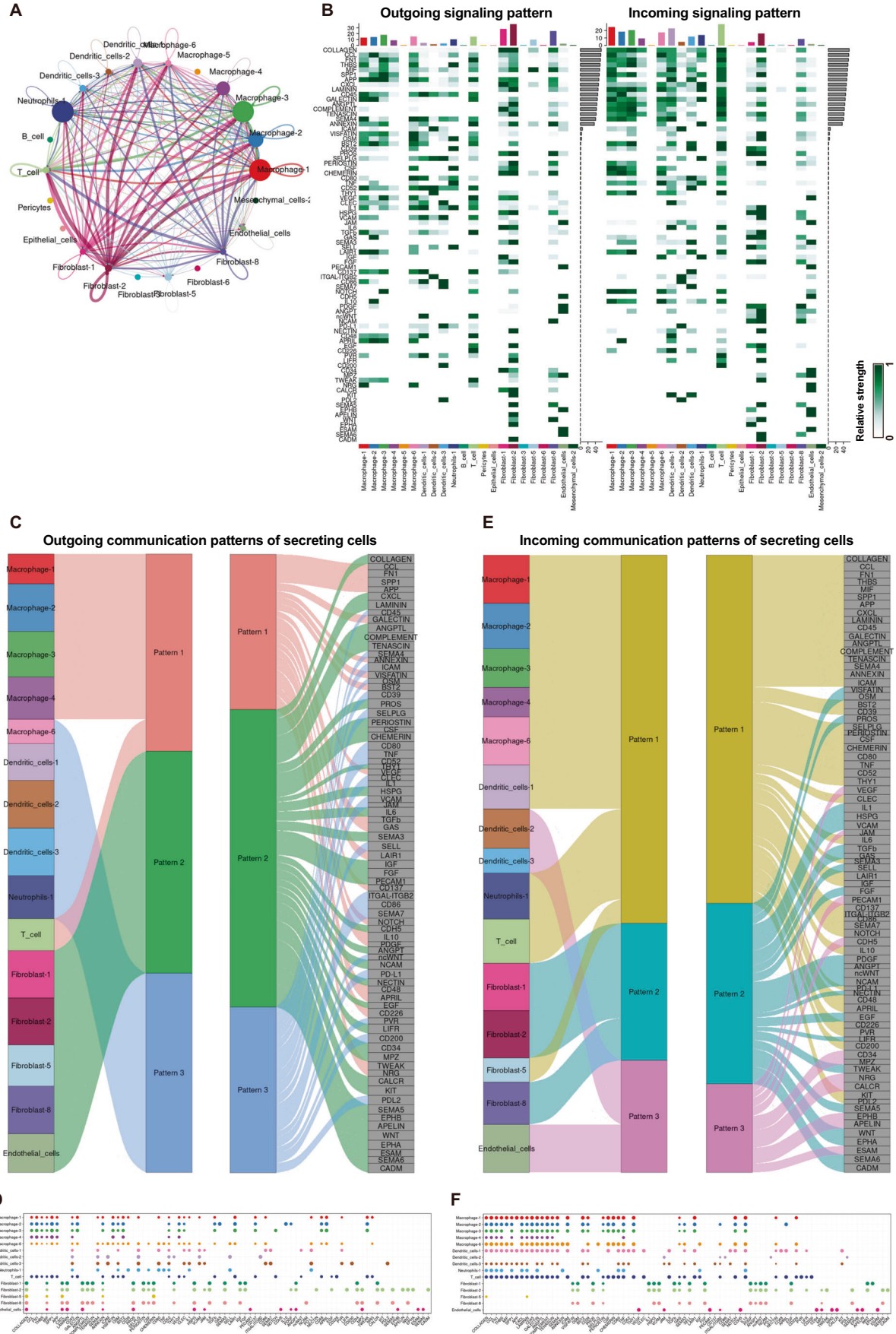

**Figure EV3. Inference with the cell–cell communication network on Day 3 post injury.**

(A) Circle plot of signaling pathways. (B) Heatmap analysis of the roles of the representative signaling pathways in the aggregated cell–cell communication network. (C) Alluvial plot of the outgoing signaling patterns of secreting cells, demonstrating the correspondence between the inferred latent pattern and cell groups, as well as the signaling pathways. The thickness of the flow indicates the contribution of the cell group or signaling pathway to each latent pattern. The height of each pattern is proportional to the number of its associated cell groups or signaling pathways. Outgoing patterns reveal how the sender cells coordinate with each other, as well as how they coordinate with certain signaling pathways to drive communications. (D) Bobble plot of the outgoing signaling patterns of secreting cells. (E) Incoming signaling patterns of target cells, showing how the target cells coordinate with each other and with certain signaling pathways to respond to incoming signals. (F) Bobble plot of the incoming signaling patterns of target cells.

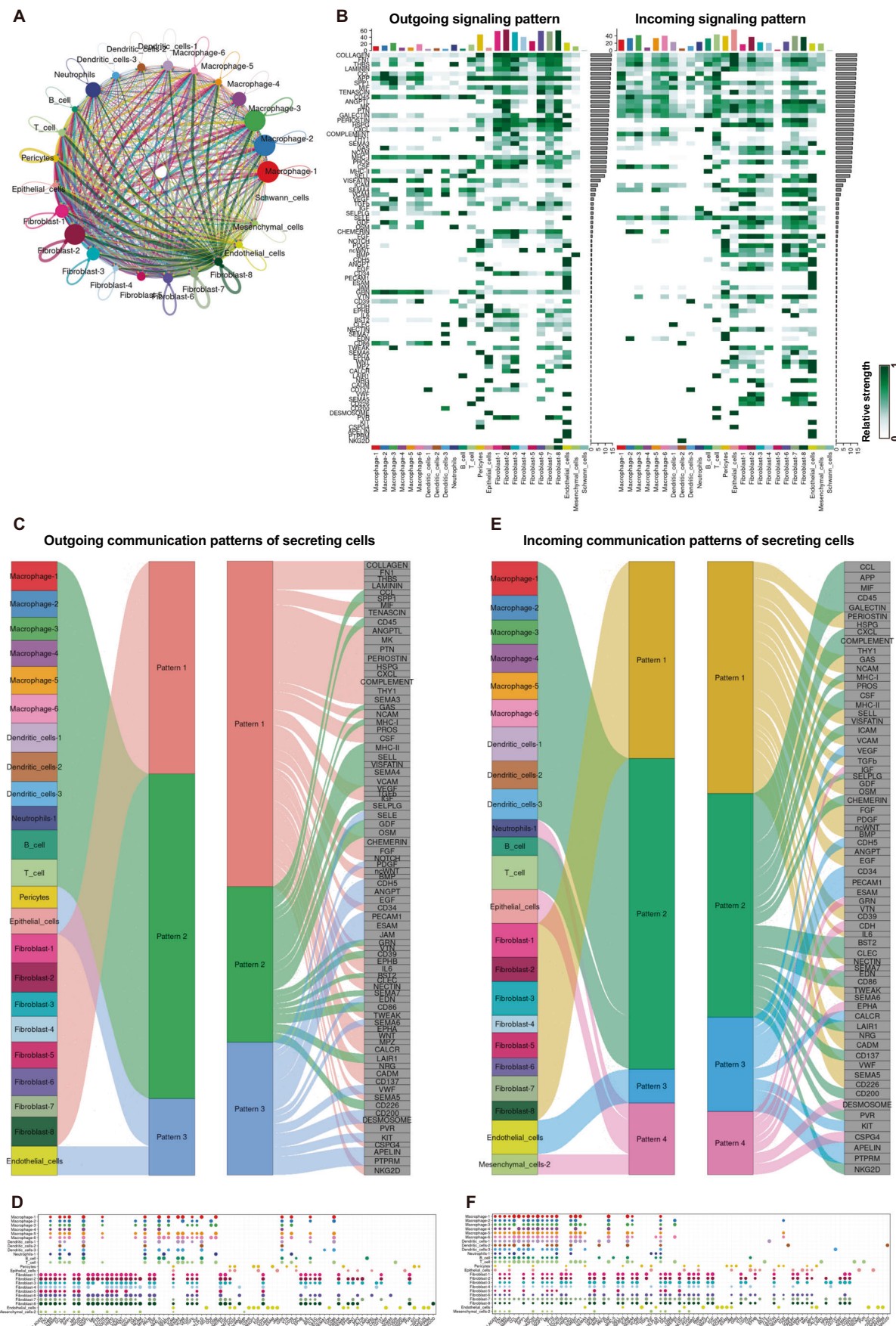

**Figure EV4.   Inference with the cell–cell communication network on Day 7 post injury.**

(A) Circle plot of signaling pathways. (B) Heatmap analysis of the roles of the representative signaling pathways in the aggregated cell–cell communication network. (C) Alluvial plot of the outgoing signaling patterns of secreting cells, demonstrating the correspondence between the inferred latent pattern and cell groups, as well as the signaling pathways. The thickness of the flow indicates the contribution of the cell group or signaling pathway to each latent pattern. The height of each pattern is proportional to the number of its associated cell groups or signaling pathways. Outgoing patterns reveal how the sender cells coordinate with each other, as well as how they coordinate with certain signaling pathways to drive communications. (D) Bobble plot of the outgoing signaling patterns of secreting cells. (E) Incoming signaling patterns of target cells, showing how the target cells coordinate with each other and with certain signaling pathways to respond to incoming signals. (F) Bobble plot of the incoming signaling patterns of target cells.

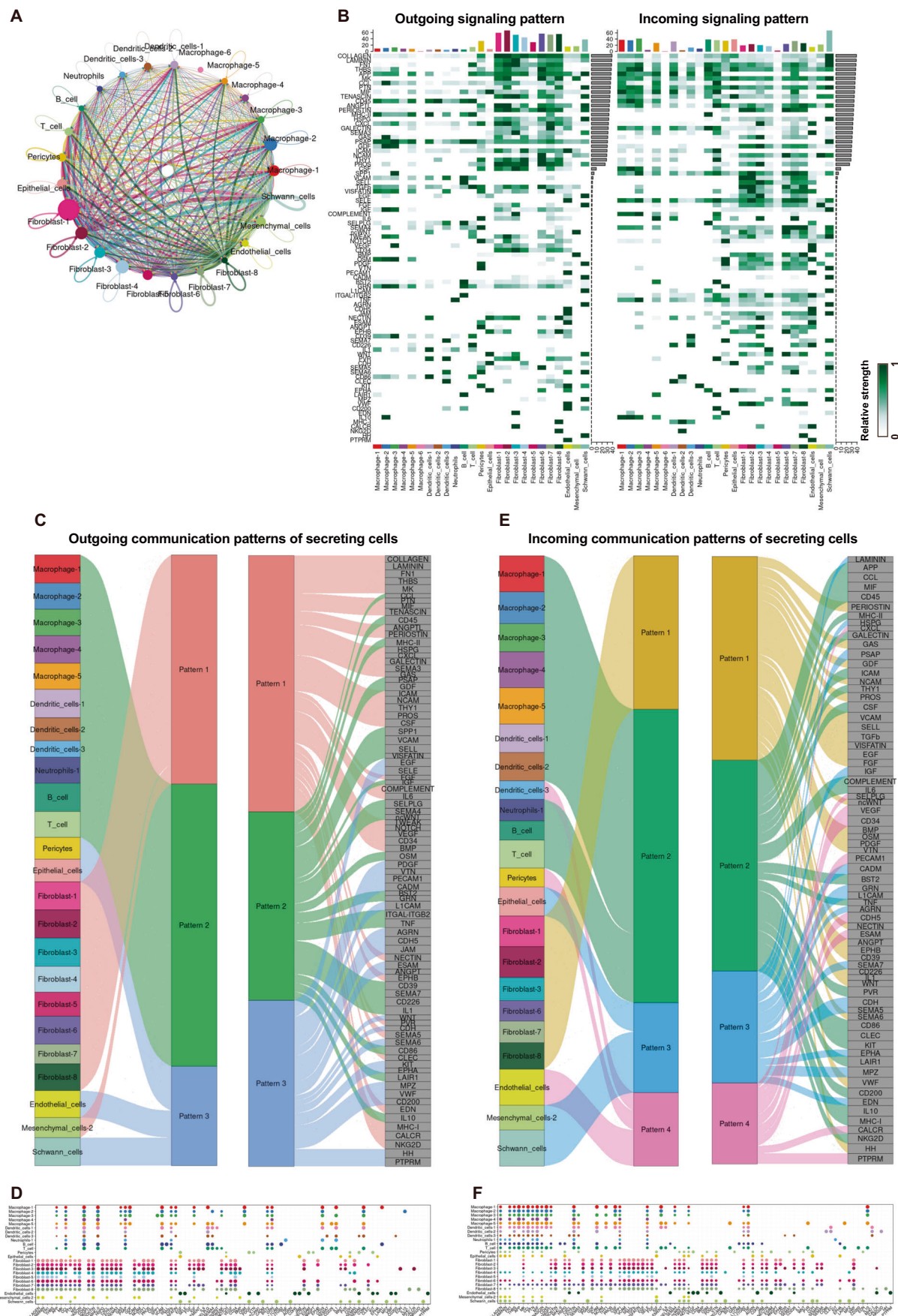

◀ **Figure EV5.  Inference with the cell–cell communication network on Day 14 post injury.**

(A) Circle plot of signaling pathways. (B) Heatmap analysis of the roles of the representative signaling pathways in the aggregated cell–cell communication network. (C) Alluvial plot of the outgoing signaling patterns of secreting cells, demonstrating the correspondence between the inferred latent pattern and cell groups, as well as the signaling pathways. The thickness of the flow indicates the contribution of the cell group or signaling pathway to each latent pattern. The height of each pattern is proportional to the number of its associated cell groups or signaling pathways. Outgoing patterns reveal how the sender cells coordinate with each other, as well as how they coordinate with certain signaling pathways to drive communications. (D) Bobble plot of the outgoing signaling patterns of secreting cells. (E) Incoming signaling patterns of target cells, showing how the target cells coordinate with each other and with certain signaling pathways to respond to incoming signals. (F) Bobble plot of the incoming signaling patterns of target cells.

