## [Peer Review File · EMBO Reports]

Novel integrated multiomics analysis reveals a key role for integrin beta-like 1 in wound scarring

Sang-Eun Kim, Ryota Noda, Yu-Chen Liu, Yukari Nakajima, Shoichiro Kameoka, Daisuke Motooka, Seiya Mizuno, Satoru Takahashi, Kento Takaya, Takehiko Murase, Kazuya Ikematsu, Katsiaryna Tratsiakova, Takahiro Motoyama, Masahiro Nakashima, Kazuo Kishi, Paul Martin, Shigeto Seno, Daisuke Okuzaki, and Ryoichi Mori

Corresponding author(s): Ryoichi Mori (ryoichi@nagasaki-u.ac.jp), Daisuke Okuzaki (dokuzaki@biken.osaka-u.ac.jp), Shigeto Seno (senoo@ist.osaka-u.ac.jp)

Review Timeline:

Submission Date:	11th Jan 24
Editorial Decision:	11th Mar 24
Revision Received:	6th Aug 24
Editorial Decision:	16th Sep 24
Revision Received:	30th Sep 24
Accepted:	24th Oct 24

Editor: Deniz Senyilmaz Tiebe

Transaction Report:

Dear Dr. Mori,

Thank you for transferring your research manuscript to our journal, which was now seen by three referees, whose reports are copied below.

My apologies for this unusual delay in getting back to you. It took longer than anticipated to receive the full set of referee reports.

Referees express interest in the analysis and the proposed role of integrin beta-like 1 in scarring. However, they also raise significant concerns that need to be addressed to consider publication here.

Given these positive recommendations, we would like to invite you to submit a revised manuscript. Please revise your manuscript with the understanding that the referee concerns (as in their reports) must be fully addressed and their suggestions taken on board. Please address all referee concerns in a complete point-by-point response. Acceptance of the manuscript will depend on a positive outcome of a second round of review. It is EMBO reports policy to allow a single round of major experimental revision only and acceptance or rejection of the manuscript will therefore depend on the completeness of your responses included in the next, final version of the manuscript.

We realize that it is difficult to revise to a specific deadline. In the interest of protecting the conceptual advance provided by the work, we recommend a revision within 3 months. Please discuss the revision progress ahead of this time with me if you require more time to complete the revisions, or if you have questions or comments regarding the revision (also by video chat).

1. A data availability section providing access to data deposited in public databases is missing (where applicable).
2. Your manuscript contains statistics and error bars based on $n=2$. Please use scatter plots in these cases.

You can submit the revision either as a Scientific Report or as a Research Article. For Scientific Reports, the revised manuscript can contain up to 5 main figures and 5 Expanded View figures, and it should not exceed 27000 characters. If the revision leads to a manuscript with more than 5 main figures it will be published as a Research Article. In this case the Results and Discussion section should be separate. If a Scientific Report is submitted, these sections have to be combined. This will help to shorten the manuscript text by eliminating some redundancy that is inevitable when discussing the same experiments twice. In either case, all materials and methods should be included in the main manuscript file.

4) a .docx formatted letter INCLUDING the reviewers' reports and your detailed point-by-point responses to their comments. As part of the EMBO publication's Transparent Editorial Process, EMBO reports publishes online a Review Process File (RPF) to accompany accepted manuscripts. This File will be published in conjunction with your paper and will include the referee reports, your point-by-point response and all pertinent correspondence relating to the manuscript. <https://www.embopress.org/page/journal/14693178/authorguide#transparentprocess>

5) a complete author checklist, which you can download from our author guidelines <https://www.embopress.org/page/journal/14693178/authorguide>. Please insert information in the checklist that is also reflected in the manuscript. The completed author checklist will also be part of the RPF.

6) Please note that all corresponding authors are required to supply an ORCID ID for their name upon submission of a revised manuscript (<<https://orcid.org/>>). Please find instructions on how to link your ORCID ID to your account in our manuscript tracking system in our Author guidelines <<https://www.embopress.org/page/journal/14693178/authorguide#authorshipguidelines>>

7) Before submitting your revision, primary datasets produced in this study need to be deposited in an appropriate public database (see <https://www.embopress.org/page/journal/14693178/authorguide#datadeposition>). Please remember to provide a reviewer password if the datasets are not yet public. The accession numbers and database should be listed in a formal "Data Availability" section placed after Materials & Method (see also <https://www.embopress.org/page/journal/14693178/authorguide#datadeposition>). Please note that the Data Availability Section is restricted to new primary data that are part of this study. * Note - All links should resolve to a page where the data can be accessed. *
If your study has not produced novel datasets, please mention this fact in the Data Availability Section.

Additional information on source data and instruction on how to label the files are available:
<https://www.embopress.org/page/journal/14693178/authorguide#sourcedata>

9) Our journal encourages inclusion of *data citations in the reference list* to directly cite datasets that were re-used and obtained from public databases. Data citations in the article text are distinct from normal bibliographical citations and should directly link to the database records from which the data can be accessed. In the main text, data citations are formatted as follows: "Data ref: Smith et al, 2001" or "Data ref: NCBI Sequence Read Archive PRJNA342805, 2017". In the Reference list, data citations must be labeled with "[DATASET]". A data reference must provide the database name, accession number/identifiers and a resolvable link to the landing page from which the data can be accessed at the end of the reference. Further instructions are available at <http://www.embopress.org/page/journal/14693178/authorguide#referencesformat>

- the name of the statistical test used to generate error bars and P values,
- the number (n) of independent experiments (please specify technical or biological replicates) underlying each data point,
- the nature of the bars and error bars (s.d., s.e.m.),
- If the data are obtained from n Program fragment delivered error `Can't locate object method "less" via package "than" (perhaps you forgot to load "than"?) at //ejpvfs23/sites23b/embor_www/letters/embor_decision_revise_and_review.txt line 56.' 2, use scatter blots showing the individual data points.

12) Please also note our reference format:

I look forward to seeing a revised version of your manuscript when it is ready. Please let me know if you have questions or comments regarding the revision.

Kind regards,

Deniz Senyilmaz Tiebe

Deniz Senyilmaz Tiebe, PhD
Scientific Editor
EMBO Reports

Referee #1:

This is an interesting study that provides novel information on the mechanism involved in scar formation during wound healing and identifies a new player in this process. The study is generally well performed and the data are convincing. The RNA-seq data are very useful for the wound healing community. However, there are also some problems with the manuscript as listed below. In particular, the IL-1b data are not convincing.

- 1.) Abstract: The authors do not show any data on the effect of TGF- β or IL-1b blocking on Itgb1 expression - therefore, the sentence in the abstract (lines 14-16) is not correct.
- 2.) It is not correct to state that spatial and scRNA-seq have not been combined to study wound healing (see introduction). In fact, this was published by the Longaker lab (Foster et al., PNAS 2021).
- 3.) The last paragraph of the introduction is another summary and could be significantly shortened.
- 4.) An important role of ITGBL1 in scar formation has previously been demonstrated (Zhao et al., Dev Cell 2023). These data should be mentioned and the study should be cited.
- 5.) Results, line 92: S100a9 is not specific for neutrophils - it is also strongly expressed by wound keratinocytes. This should be considered - please comment. Krt10 is only expressed by a subset of (differentiated) wound keratinocytes - most wound keratinocytes express Krt14. Finally, fibroblasts and pericytes are also mesenchymal cells and BMP4 is not a specific marker for mesenchymal cells.
- 6.) Line 108: The number of genes in fibroblasts remains of course unaltered - please change to; "the number of expressed genes".
- 7.) Line 146: It has previously been shown that wound macrophages proliferate in the wound tissue (see for example Pang et al., 2020). This should be mentioned in this context.
- 8.) The last sentence on page 11 should be deleted - this statement is highly speculative and cannot be concluded from the RNA-seq data.
- 9.) It is not clear why the authors used excisional wounds for the scRNA-seq analysis, but then switched to incisional wounds for the comparison of wt and PU.1 ko mice. It would have been better to use the same wound healing model for all studies to allow a direct comparison.
- 10.) The authors identified 1057 genes that are downregulated in wounds of PU.1 ko mice - are there also genes that are upregulated?
- 11.) Page 15, line 285: please explain what you mean with "pathological regions".
- 12.) The authors should report on the phenotype of the Itgb1 ko in non-injured mice. In particular, it is important to know if the mice have general health problems that may affect the wound healing process and if the non-wounded skin is already affected.
- 13.) The data shown in Fig. 5H are very important, but unfortunately not convincing. There is only a difference at one time point (day 7) and the variability of obviously very high. This result should be confirmed by histomorphometric analysis of the wounds - at least at day 7. Is there a difference in wound contraction or in wound reepithelialisation or both?
- 14.) The Picosirius Red stainings look convincing (Fig. 5I), but the data should be quantified. Is the area of scar tissue reduced? Looking at the arrowheads in Fig. 5I, it seems even enlarged. This should be clarified. Is there also less collagen in general - this should be checked by hydroxyproline analysis.
- 15.) The results shown in Fig. 6A are not surprising, because it has previously been shown that these genes are regulated by TGF- β . This figure could be moved to the Supplement.
- 16.) Fig. 6D: IL-1b has obviously no effect on Itgb1 expression in mouse fibroblasts - this should be clearly stated. In addition, the difference seen with IL-1b in the TGF- β treated fibroblasts (Fig. 6E) is very minor - these data are not convincing. They should be repeated or removed. There is clearly insufficient evidence for the statement made in this context (lines 403-406).
- 17.) The authors have no information on the levels of IL-1b protein in wounds - therefore, it is unclear if the concentrations used for the experiments in Fig. 6 are comparable to those seen at the wound site.

- 18.) Is the high concentration of IL-1b simply toxic for human fibroblasts? This should be clarified. Does IL-1b have an effect on TGF- β -treated human fibroblasts? This experiment (done with mouse fibroblasts, see Fig. 6E) should also be done with human fibroblasts. In summary, there is insufficient evidence for the model shown in Fig. 6G. I suggest removing the IL-1b data.
- 19.) Fig. 7C is of low quality and it is impossible to see the double positive cells.
- 20.) It is very interesting that the authors also show delayed wound healing mice lacking *Itgbl1* only in myofibroblasts. However, the specificity of the knockout in this cell type is not shown. In addition, the difference was again only seen at one time point. Please show histomorphometric data and compare them to histomorphometric data of the global ko mice.
- 21.) The authors cannot conclude that expression of *Itgbl1* in myofibroblasts is indispensable for granulation tissue formation. First, there was only a difference in healing at one time and the wounds clearly still heal and second, the granulation tissue of these wounds is not shown (see comment 17 regarding histomorphometry).
- 22.) The discussion is too long and includes speculations that are not supported by the data shown in this manuscript.
- 23.) The manuscript includes several errors in spelling and grammar, which should be corrected.

Referee #2:

This is an interesting and extensive mouse study on wound healing. The authors utilize novel techniques e.g. integrated spatial multiomics and identify a distinct cell population (*acta2* highly positive myofibroblasts) that is important in tissue regeneration. I congratulate the authors on this extensive and well-done study.

Minor points:

- The experiments are mainly done in mice. Could the authors comment on wound healing in humans? Are the *acta2* highly positive myofibroblasts in human skin or a corresponding cell population? Can this mouse study translate into the pathophysiology of tissue regeneration in patients?
- Typo on page 18, line 363
- The manuscript is rather long. Please shorten the text.

Referee #3:

In this article the authors have explored the cell types and the molecular mechanisms that govern the inflammation driven scarring of the skin. Skin wound healing is a process involving inflammatory phase, proliferation phase and maturation phase. Lot happens during these phases starting with inflammatory cell infiltration, re-epithelialization and formation of granulation tissue and finally degradation of extracellular matrix. Wound healing is a complex process with actively involving cellular and molecular level changes in the wound microenvironment, but what happens during inflammation driven scarring, remains unclear. To address this authors have developed very interesting novel strategy, by combining high resolution spatial multiomics methods integrating spatial transcriptome utilizing 10 x Genomics Visium platform with temporal scRNA-seq datasets to identify new cell-cell communications and signaling during the wound repair process. Single cell RNA-Seq on murine skin wounds provided information about the temporal dynamic changes in the single cell populations and genes expressed in the wound. Also authors studied the gene expression heterogeneity in macrophages and fibroblasts involved in the wound healing. They have studied the cell-cell communications by cellchat at the skin wound sites. For Spatial transcriptional expression of inflammation related genes authors have utilized the Visium platform for 697 tissue spots at the murine wound site, which are data points equivalent to microdissection. The data generated from these two methods and PU.1^{-/-} mice that lack inflammatory response was combined and screened to identify the candidate genes involved in inflammation and scarring during skin wound. These experiments led to identification of nine genes with potential involvement in inflammation-related scarring, including integrin beta-like 1 (*Itgbl1*) that is regulated by antagonistic interplay between TGF β 1 and IL1 β in the dermal fibroblasts. Further CRISPR-Cas technology to generate transgenic mice, authors have confirmed the role of *Itgbl1* in skin wound healing and regulation of fibrogenesis and the *Itgbl1*-expressing cells were involved in collagen organization and are a pivotal part of the initiation of scarring at skin wound sites. Also, minority population of *Acta2*^{high}-expressing myofibroblasts appears to be involved in scarring in association with *Itgbl1* expression. Overall, the new methodology and data presented by authors will help paving the way towards deeper understanding of skin wound healing and will help in development of new tools for clinical applications to study inflammation and fibrosis.

Major comments: The authors have integrated the spatial and temporal gene expression profiles and have shortlisted the potential genes involved in inflammation related scarring during the wound healing. The authors conducted vigorous studies and have provided lot of information. There are major concerns that need to be addressed:

- 1) Authors have discussed the gene expression changes in macrophages but have not detailed the crosstalk between the macrophage and the fibroblasts during early stages of wound healing. What are the macrophage-to-fibroblast interaction pathways that are involved in fibroblast activation? How macrophages are involved in the activation of fibroblasts. Did authors performed any experiments involving inhibiting TGF β 1 signaling and check its effect on fibroblast activation mediated by

macrophages.

2) Did author's study change in expression levels of type I collagen gene (Col1a1) using the Visium platform at day 7 and 14? After IL1 β treatment, was there any increase in the levels of collagen 3A1. Was there any effect on the ratio of collagen 1/collagen 3. Did author's confirmed IL1 β levels at later time points like day 14?

Minor comments:

1) Please revise the manuscript to correct the typing errors; Page 4, Line 40, change reveled to revealed. Figure 7L- please correct assenbly to assembly. Page 18 line 363-its an incomplete sentence.

2) Please specify in methods section, how many mice (biological and technical replicates) were utilized for scRNA-Seq and Visium platform. Are 40,000 cells from a wound sufficient to get to a conclusion, more sample size will help to address this question.

Professor Bernd Pulverer
Editor-in-Chief
EMBO Reports

Re: Manuscript Number: EMBOR-2024-58791-T

6 August 2024

Dear Professor Pulverer,

Please find attached our paper entitled “Novel integrated multiomics analysis reveals a key role for integrin beta-like 1 in wound scarring”, by Sang-Eun Kim et al., which we would like you to reconsider for publication as a Research Article in *EMBO Reports*.

We are grateful for the positive comments from the referees and the editor. Please find below our responses to all comments (changes to the text in the revised manuscript are indicated by red font). We hope that the revised manuscript is now suitable for publication in *EMBO Reports*.

We look forward to hearing from you at your earliest convenience.

Yours sincerely,

Ryoichi Mori

Editor comments:

11 March, 2024

Dear Dr. Mori,

Thank you for transferring your research manuscript to our journal, which was now seen by three referees, whose reports are copied below.

My apologies for this unusual delay in getting back to you. It took longer than anticipated to receive the full set of referee reports.

Referees express interest in the analysis and the proposed role of integrin beta-like 1 in scarring. However, they also raise significant concerns that need to be addressed to consider publication here.

Given these positive recommendations, we would like to invite you to submit a revised manuscript. Please revise your manuscript with the understanding that the referee concerns (as in their reports) must be fully addressed and their suggestions taken on board. Please address all referee concerns in a complete point-by-point response. Acceptance of the manuscript will depend on a positive outcome of a second round of review. It is EMBO reports policy to allow a single round of major experimental revision only and acceptance or rejection of the manuscript will therefore depend on the completeness of your responses included in the next, final version of the manuscript.

Comment 1: *We realize that it is difficult to revise to a specific deadline. In the interest of protecting the conceptual advance provided by the work, we recommend a revision within 3 months. Please discuss the revision progress ahead of this time with me if you require more time to complete the revisions, or if you have questions or comments regarding the revision (also by video chat).*

Response: Thank you very much for extending the deadline.

Comment 2: *IMPORTANT NOTE: we perform an initial quality control of all revised manuscripts before re-review. Your manuscript will FAIL this control and the handling will be DELAYED if the following APPLIES:*

- 1. A data availability section providing access to data deposited in public databases is missing (where applicable).*
- 2. Your manuscript contains statistics and error bars based on $n=2$. Please use scatter plots in these cases.*

Response: The reported Visium, scRNA-Seq, and mRNA-Seq data are available in the NCBI Gene Expression Omnibus under accession number GSE234272 (page 38, lines 817). Scatter plots were used for data visualization.

Comment 3: *You can submit the revision either as a Scientific Report or as a Research Article. For Scientific Reports, the revised manuscript can contain up to 5 main figures and 5 Expanded View figures, and it should not exceed 27000 characters. If the revision leads to a manuscript with more than 5 main figures it will be published as a Research Article. In this case the Results and Discussion section should be separate. If a Scientific Report is submitted, these sections have to be combined. This will help to shorten the manuscript text by eliminating some redundancy that is inevitable when discussing the same experiments twice. In either case, all materials and methods should be included in the main manuscript file.*

Response: We would like to submit our manuscript as a Research Article containing 7 figures, and 5 Expanded View figures.

Comment 4: *When submitting your revised manuscript, please carefully review the instructions that follow below. Failure to include requested items will delay the evaluation of your revision.*

Response: Changes made to the text in the resubmitted manuscript are indicated in red.

Comment 5: *2) individual production quality figure files as .eps, .tif, .jpg (one file per figure). See https://wol-prod-cdn.literatumonline.com/pb-assets/embo-site/EMBOPress_Figure_Guidelines_061115-1561436025777.pdf for more info on how to prepare your figures.*

Response: Individual figure files are saved in EPS format.

Comment 6: *3) We replaced Supplementary Information with Expanded View (EV) Figures and Tables that are collapsible/expandable online. A maximum of 5 EV Figures can be typeset. EV Figures should be cited as 'Figure EV1, Figure EV2' etc... in the text and their respective legends should be included in the main text after the legends of regular figures.*

*- For the figures that you do NOT wish to display as Expanded View figures, they should be bundled together with their legends in a single PDF file called *Appendix*, which should start with a short Table of Content. Appendix figures should be referred to in the main text as:*

"Appendix Figure S1, Appendix Figure S2" etc. See detailed instructions regarding expanded view here:

<https://www.embopress.org/page/journal/14693178/authorguide#expandedview>;

Response: We have made the necessary preparations based on your comments.

Comment 7: 4) a .docx formatted letter INCLUDING the reviewers' reports and your detailed point-by-point responses to their comments. As part of the EMBO publication's Transparent Editorial Process, EMBO reports publishes online a Review Process File (RPF) to accompany accepted manuscripts. This File will be published in conjunction with your paper and will include the referee reports, your point-by-point response and all pertinent correspondence relating to the manuscript.

<https://www.embopress.org/page/journal/14693178/authorguide#transparentprocess>

Response: We understand the EMBO Publications Transparent Editorial Process initiative and agree to publication of the RPF with our manuscript.

Comment 8: 5) a complete author checklist, which you can download from our author guidelines <https://www.embopress.org/page/journal/14693178/authorguide>. Please insert information in the checklist that is also reflected in the manuscript. The completed author checklist will also be part of the RPF.

Response: We have submitted the checklist as requested.

Comment 9: 6) Please note that all corresponding authors are required to supply an ORCID ID for their name upon submission of a revised manuscript (<https://orcid.org/>;). Please find instructions on how to link your ORCID ID to your account in our manuscript tracking system in our Author guidelines

<https://www.embopress.org/page/journal/14693178/authorguide#authorshipguidelines>;

Response: We have included the ORCID ID for the first author (SEK: 0000-0002-5811-759X) and all corresponding authors (SS: 0000-0003-3861-6444, DO: 0000-0002-4552-783X, and RM: 0000-0002-7596-9620).

Comment 10: 7) *Before submitting your revision, primary datasets produced in this study need to be deposited in an appropriate public database (see <https://www.embopress.org/page/journal/14693178/authorguide#datadeposition>). Please remember to provide a reviewer password if the datasets are not yet public. The accession numbers and database should be listed in a formal "Data Availability" section placed after Materials & Method (see also <https://www.embopress.org/page/journal/14693178/authorguide#datadeposition>). Please note that the Data Availability Section is restricted to new primary data that are part of this study. * Note - All links should resolve to a page where the data can be accessed. **

Response: The reported Visium, scRNA-Seq, and mRNA-Seq datasets are available in the NCBI Gene Expression Omnibus under accession no. GSE234272 (p38, line 817).

Comment 11: 8) *At EMBO Press we ask authors to provide source data for the main figures. Our source data coordinator will contact you to discuss which figure panels we would need source data for and will also provide you with helpful tips on how to upload and organize the files.*

Additional information on source data and instruction on how to label the files are available: <https://www.embopress.org/page/journal/14693178/authorguide#sourcedata>

Response: We have uploaded the SourceData for the main figures. However, we would like to send the SourceData for the Fig 5 and Fig7 to the editorial office because the file size are too large to upload.

Comment 12: 9) *Our journal encourages inclusion of *data citations in the reference list* to directly cite datasets that were re-used and obtained from public databases. Data citations in the article text are distinct from normal bibliographical citations and should directly link to the database records from which the data can be accessed. In the main text, data citations are formatted as follows: "Data ref: Smith et al, 2001" or "Data ref: NCBI Sequence Read Archive PRJNA342805, 2017". In the Reference list, data citations must be labeled with "[DATASET]". A data reference must provide the database name, accession*

number/identifiers and a resolvable link to the landing page from which the data can be accessed at the end of the reference. Further instructions are available at <http://www.embopress.org/page/journal/14693178/authorguide#referencesformat>

Response: We did not use publicly available datasets in this study.

Comment 13: 10) Regarding data quantification (see Figure Legends: <https://www.embopress.org/page/journal/14693178/authorguide#figureformat>)

Response: We have modified the figure legends as indicated.

Comment 14: 11) The journal requires a statement specifying whether or not authors have competing interests (defined as all potential or actual interests that could be perceived to influence the presentation or interpretation of an article). In case of competing interests, this must be specified in your disclosure statement. Further information: <https://www.embopress.org/competing-interests>

Response: The authors declare no competing interests.

Comment 15: 12) Please also note our reference format:

Response: We have checked and corrected the reference format throughout the manuscript.

Comment 16: We would also welcome the submission of cover suggestions, or motifs to be used by our Graphics Illustrator in designing a cover.

I look forward to seeing a revised version of your manuscript when it is ready. Please let me know if you have questions or comments regarding the revision.

Response: Ryoichi Mori, the corresponding author, has changed his affiliation to the Department of Tissue Repair and Regenerative Medical Science, Atomic Bomb Disease Institute, Nagasaki University. His contribution now includes responding to referee comments. Tratsiakova, Takahiro Motoyama, and Masahiro Nakajima performed histological analysis to revise this manuscript, so we added them as co-authors. We have updated this information in the manuscript (pages 1 and 2).

Kind regards,

Deniz Senyilmaz Tiebe

Deniz Senyilmaz Tiebe, PhD

Scientific Editor

EMBO Reports

Referee comments:

Referee #1:

This is an interesting study that provides novel information on the mechanism involved in scar formation during wound healing and identifies a new player in this process. The study is generally well performed and the data are convincing. The RNA-seq data are very useful for the wound healing community. However, there are also some problems with the manuscript as listed below. In particular, the IL-1b data are not convincing.

Comment 1: 1.) *Abstract: The authors do not show any data on the effect of TGF- β or IL-1b blocking on *Itgbl1* expression - therefore, the sentence in the abstract (lines 14-16) is not correct.*

Response: In accordance with your comments, we have modified the sentence regarding the effect of TGF β 1 and IL1 β on *Itgbl1* expression (page 3, lines 14 to 15).

Comment 2: 2.) *It is not correct to state that spatial and scRNA-seq have not been combined to study wound healing (see introduction). In fact, this was published by the Longaker lab (Foster et al., PNAS 2021).*

Response: As you mentioned, studies integrating Visium and scRNA-seq have been reported. We believe that Foster and colleagues' study used scRNA-seq data as an intermediary to integrate Visium and scATAC-seq data, but it was limited to focusing on fibroblasts and excluded contributions from non-fibroblasts. In our study, we conducted cell-type deconvolution and cell-cell communication analysis for signal direction (COMMOT) through integrated analysis. We have incorporated these additional advantages into the text (page 5, lines 55 to 57).

Comment 3: 3.) *The last paragraph of the introduction is another summary and could be significantly shortened.*

Response: As requested, we have modified the last paragraph of the introduction, reducing it from 159 words to 123 words (page 5 to 6, lines 61 to 71).

Comment 4: 4.) *An important role of *ITGBL1* in scar formation has previously been demonstrated (Zhao et al., Dev Cell 2023). These data should be mentioned and the study should be cited.*

Response: We have modified the sentence as indicated (page 17, lines 344 to 346).

Comment 5: 5.) *Results, line 92: S100a9 is not specific for neutrophils - it is also strongly expressed by wound keratinocytes. This should be considered - please comment. Krt10 is only expressed by a subset of (differentiated) wound keratinocytes - most wound keratinocytes express Krt14. Finally, fibroblasts and pericytes are also mesenchymal cells and BMP4 is not a specific marker for mesenchymal cells.*

Response: We acknowledge that our wording "cell-specific marker" in the manuscript may have been misleading or unclear. The reason for using cell-specific markers is to indicate representative genes identified as differentially expressed within clusters of our scRNA-seq data using the "find-marker" function in BBrowser (Dataset EV2). We initially used the cell marker gene search function in the BBrowser database to estimate broad cell population classifications based on our single-cell analysis results. Then, based on those results (Dataset EV2), we presented the top and literature-reported genes as representatives in Appendix Figure S1. Therefore, the cell-specific markers used in this paper may be markers inferred only for murine skin wounds. Each cell type can have several markers, and these can vary depending on the organ. Additionally, marker genes may not be included depending on the single-cell analysis results. This analysis is a screening conducted for the initial estimation of the cell population (i.e., semi-automated cell-type annotation, see page 6, lines 86 to 92). The identification of precise cell-type markers for each cell was carried out in the subsequent sub-clustering (Figure 1B and Dataset EV2).

Comment 6: 6.) *Line 108: The number of genes in fibroblasts remains of course unaltered - please change to; "the number of expressed genes".*

Response: We have modified the sentence as indicated (page 7, line 106).

Comment 7: 7.) *Line 146: It has previously been shown that wound macrophages proliferate in the wound tissue (see for example Pang et al., 2020). This should be mentioned in this context.*

Response: Thank you for your suggestion. We have added a sentence including this reference (Pang J, Leukoc Biol, 2020, PMID: 31777992) (page 9, lines 139).

Comment 8: 8.) *The last sentence on page 11 should be deleted - this statement is highly speculative and cannot be concluded from the RNA-seq data.*

Response: We have deleted the sentence as indicated (page 11, line 200 to 202).

Comment 9: 9.) *It is not clear why the authors used excisional wounds for the scRNA-seq analysis, but then switched to incisional wounds for the comparison of wt and PU.1 ko mice. It would have been better to use the same wound healing model for all studies to allow a direct comparison.*

Response: If antibiotics are not administered, *PU.1*^{-/-} mice will die within approximately 2-3 days, so only neonatal mice can be used. The creation of excisional wounds using a biopsy punch in neonatal mice is difficult, so we have been using incisional wounds for analysis with *PU.1*^{-/-} mice instead (Martin P et al, *Curr Biol*, 2003, PMID: 12842011) (de Kerckhove M et al, *EMBO Mol Med*, 2018, PMID: 30171089). Considering our analysis results, we believe that gene expression dynamics in incisional wounds after 24 hours are more similar to the proliferation phase than the inflammatory phase. Therefore, we conducted functional analysis using 24-hour incisional wound samples in this study.

Comment 10: 10.) *The authors identified 1057 genes that are downregulated in wounds of PU.1 ko mice - are there also genes that are upregulated?*

Response: We identified 565 genes that are upregulated (with a cutoff of < 2) in the wounds of *PU.1*^{-/-} mice. The gene expression data from our comparative study using *PU.1*^{-/-} and WT mice are shown in Dataset EV4.

Comment 11: 11.) *Page 15, line 285: please explain what you mean with "pathological regions".*

Response: We apologize for the erroneous word "pathological." We have revised it to "wound" (page 14, line 281).

Comment 12: 12.) *The authors should report on the phenotype of the Itgb1 ko in non-injured mice. In particular, it is important to know if the mice have general health problems that may affect the wound healing process and if the non-wounded skin is already affected.*

Response: There are no apparent changes in the appearance of young *Itgb1*^{-/-} mice (8 weeks old), and no problems have been observed during adolescence (see below Fig 1 for referee only). Even when *Itgb1*^{-/-} mice are mated with each other, they can give birth without any problems. However, considering that *Itgb1* is reportedly associated with cancer development, we plan to comprehensively investigate its functional role in the aging process during future studies. We have added this information to the text (page 18, line 361 to 362).

Figure 1 (referee only). No phenotypic abnormalities in young *Itgbl1*^{-/-} mice.

Representative images of the gross appearance of WT and *Itgbl1*^{-/-} mice. One side is 1 cm square.

Comment 13: 13.) *The data shown in Fig. 5H are very important, but unfortunately not convincing. There is only a difference at one time point (day 7) and the variability of obviously very high. This result should be confirmed by histomorphometric analysis of the wounds - at least at day 7. Is there a difference in wound contraction or in wound reepithelialisation or both?*

Response: As suggested, we measured re-epithelialization at wound sites on day 7 post injury. Re-epithelialization on Day 7 post injury was successfully completed in both WT mice and *Itgbl1*^{-/-} mice. However, the wound area on Days 7 and 14 post injury was significantly increased in *Itgbl1*^{-/-} mice compared with WT mice. We added a sentence describing these differences. We have modified the text accordingly (page 18, line 365 to 367) and Fig 5I and J.

To test wound contraction, we investigated the expression of *Acta2* at wound sites in both groups of mice but observed no differences. The delay in gross appearance of skin wound healing only on Day 7 post injury may have occurred because *Itgbl1* supports fibrogenesis during granulation tissue formation but does not regulate collagen gene expression (page 18, line 377 to 379) (Appendix Fig S5).

Comment 14: 14.) *The Picrosirius Red stainings look convincing (Fig. 5I), but the data should be quantified. Is the area of scar tissue reduced? Looking at the arrowheads in Fig. 5I, it seems even enlarged. This should be clarified. Is there also less collagen in general - this should be checked by hydroxyproline analysis.*

Response: As suggested, we investigated the granulation tissue area on Day 14 and found that it was significantly increased in *Itgbl1*^{-/-} mice compared with WT mice (Fig 5I and J).

We performed hydroxyproline analysis and confirmed that fibrogenesis at scar sites was markedly decreased in *Itgbl1*^{-/-} mice (Fig 5M). We have modified the text accordingly (page 18, lines 375 to 377).

Comment 15: 15.) *The results shown in Fig. 6A are not surprising, because it has previously been shown that these genes are regulated by TGF- β . This figure could be moved to the Supplement.*

Response: As suggested, Fig 6A has been moved to Appendix Fig S7B.

Comment 16: 16.) *Fig. 6D: IL-1 β has obviously no effect on Itgbl1 expression in mouse fibroblasts - this should be clearly stated. In addition, the difference seen with IL-1 β in the TGF- β treated fibroblasts (Fig. 6E) is very minor - these data are not convincing. They should be repeated or removed. There is clearly insufficient evidence for the statement made in this context (lines 403-406).*

Response: As requested, we have added an explanation regarding the lack of effect of *Itgbl1* in murine dermal skin-derived primary fibroblasts (MDF) treated with IL1 β alone (page 20, line 408).

Regarding the difference observed with IL1 β in TGF β 1-treated MDF (Fig 6E), we repeated these experiments and confirmed that *Itgbl1* expression in TGF β 1-treated MDF was markedly attenuated by IL1 β (Fig 6D).

Comment 17: 17.) *The authors have no information on the levels of IL-1 β protein in wounds - therefore, it is unclear if the concentrations used for the experiments in Fig. 6 are comparable to those seen at the wound site.*

Response: We measured the concentration of IL1 β at intact skin and wound sites in WT mice on Days 3, 7, and 14 post injury by ELISA (Appendix Fig S3F). This result aligns with the trends observed in our scRNA-Seq data (Appendix Fig S3D).

As you mentioned regarding the concentrations used for experiments in Fig 6D and F, the IL1 β concentration for *in vitro* experiments cannot be directly compared with the ELISA results. In our *in vitro* experiments, we aimed to identify the lowest possible concentration that would elicit a response (Fig 6C and E) and determined that 100 pg/mL was appropriate.

Comment 18: 18.) *Is the high concentration of IL-1 β simply toxic for human fibroblasts? This should be clarified. Does IL-1 β have an effect on TGF- β -treated human fibroblasts? This experiment (done with mouse fibroblasts, see Fig. 6E) should also be done with human fibroblasts. In summary, there is insufficient evidence for the model shown in Fig. 6G. I suggest removing the IL-1 β data.*

Response: Our microscopic analysis showed that IL1 β treatment on MDFs and HDFs, up to 100 pg/mL, does not alter cell morphology. Therefore, we believe that 100 pg/mL of IL1 β is not toxic to primary fibroblasts from mice or humans in our *in vitro* experiments.

As suggested, we investigated the expression of *ITGBL1* in TGF β 1-treated HDF inhibited by IL1 β signaling; we found that the expression of *ITGBL1* in TGF β 1-treated HDF was significantly inhibited by IL1 β stimulation (Fig 6F). Based on these results, we would like to propose our hypothesis regarding the function of IL1 β in TGF β 1-treated fibroblasts in this continuing study.

Comment 19: 19.) *Fig. 7C is of low quality and it is impossible to see the double positive cells.*

Response: We modified Fig 7C in accordance with your comment.

Comment 20: 20.) *It is very interesting that the authors also show delayed wound healing mice lacking *Itgbl1* only in myofibroblasts. However, the specificity of the knockout in this cell type is not shown. In addition, the difference was again only seen at one time point. Please show histomorphometric data and compare them to histomorphometric data of the global ko mice.*

Response: We confirmed that *Itgbl1* mRNA expression at wound sites of *Itgbl1^{flox/flox}::Tagln^{Cre/Cre}* mice was significantly decreased compared with control mice at Day 7 post injury (Fig 7J). We believe that the specific downregulation of *Itgbl1* expression in myofibroblasts of *Itgbl1^{flox/flox}::Tagln^{Cre/Cre}* mice is responsible for the wound repair effects observed on Day 7 after injury. This assumption is supported by the following points. (i) Based on our scRNA-seq results, *Tagln* and *Acta2* are expressed in fibroblast and pericyte clusters (Fig EV2B). (ii) *Tagln* and *Acta2* are significantly expressed in F6 and F8 fibroblast subclusters (Fig EV2B), which specifically appear on Day 7 after injury and largely disappear by Day 14 post injury (Fig 1B and C). (iii) *Itgbl1* is expressed in the F6 fibroblast subcluster (Log₂FC:0.9) but not in the F8 subcluster (Log₂FC, -0.148) (Fig 2D) (Dataset EV3). However, no expression is observed in pericytes (Log₂FC: -0.526). Therefore, the significant delay observed in *Itgbl1^{flox/flox}::Tagln^{Cre/Cre}* mice on Day 7 post injury is likely due to the expression of *Itgbl1* in the F6 fibroblast subcluster.

The incomplete downregulation of *Itgbl1* in *Itgbl1^{flox/flox}::Tagln^{Cre/Cre}* mice can be attributed to the following factors. (i) It has been reported that achieving complete Cre-LoxP-based conditional knockout is inherently difficult on a purely theoretical basis, making it challenging to attain 100% knockout efficiency in the targeted cell lineage (Heffner

CS et al, *Nat Commun*, 2012, PMID: 23169059). (ii) The efficiency of NMD-mediated degradation of the floxed target gene via Cre-mediated recombination is known to depend on the cell type and the gene itself (Zetoune AB et al, *BMC Genet*, 2008, PMID: 19061508) (Lindeboom RG et al, *Nat Genet*, 2016, PMID: 27618451). However, *Tglin*-Cre transgenic mice have been widely utilized in studies specifically involving myofibroblast knockout analysis; therefore, we believe that our *Itgbl1^{flox/flox}::Tagln^{Cre/Cre}* mice constitute an appropriate model.

Based on your suggestion regarding histological analysis, we measured re-epithelialization and granulation tissue area at wound sites on Day 7 post injury by H&E staining. Re-epithelialization on Day 7 post injury was successfully completed in both control (*Itgbl1^{flox/flox}*) mice and *Itgbl1^{flox/flox}::Tagln^{Cre/Cre}* mice. However, the granulation tissue area was significantly increased in *Itgbl1^{flox/flox}::Tagln^{Cre/Cre}* mice compared with control mice (Fig 7M and N). Collectively, these results indicate that wound contraction are attenuated in *Itgbl1^{flox/flox}::Tagln^{Cre/Cre}* mice. We have modified the text accordingly (page 22, lines 467 to 468) and Fig 7M and N.

Comment 21: 21.) *The authors cannot conclude that expression of Itgbl1 in myofibroblasts is indispensable for granulation tissue formation. First, there was only a difference in healing at one time and the wounds clearly still heal and second, the granulation tissue of these wounds is not shown (see comment 17 regarding histomorphometry).*

Response: As mentioned in our response to comment 20, we believe that *Itgbl1* in *Tglin^{high}* and *Acta2^{high}* cells is essential on Day 7, when these highly expressing cells are transiently present. Although the wound eventually heals, we suspect that *Itgbl1* is essential for granulation tissue formation, considering its temporary effect.

Comment 22: 22.) *The discussion is too long and includes speculations that are not supported by the data shown in this manuscript.*

Response: We have modified the Discussion as indicated.

Comment 23: 23.) *The manuscript includes several errors in spelling and grammar, which should be corrected.*

Response: As requested, all typing and grammatical errors in the text and Figures have been checked and corrected by a professional English editing service.

Referee #2:

This is an interesting and extensive mouse study on wound healing. The authors utilize novel techniques e.g. integrated spatial multiomics and identify a distinct cell population (acta2 highly positive myofibroblasts) that is important in tissue regeneration. I congratulate the authors on this extensive and well-done study.

Comment 1: *Minor points:*

- The experiments are mainly done in mice. Could the authors comment on wound healing in humans? Are the acta2 highly positive myofibroblasts in human skin or a corresponding cell population? Can this mouse study translate into the pathophysiology of tissue regeneration in patients?

Response: It is known that many ACTA2-positive myofibroblasts exist in human skin wounds and scar sites (Desmoulière A et al, *Wound Repair Regen*, 2005, PMID: 15659031).

Our data regarding the expression of *ITGBL1* in human-dermal skin-derived primary fibroblasts (Fig 6B) suggest that *ITGBL1* is involved in keloid development. Additionally, Zhao and colleagues reported that reduced SET and MYND domain containing 3 (*SMYD3*) and *ITGBL1* expression levels are strongly associated with the anti-scarring effects of pressure therapy in clinical specimens (Zhao J et al, *Dev Cell*, 2023, PMID: 37192621). However, there are no reports concerning the function of *ITGBL1* in human skin wound healing. We have added text regarding the translational potential of our findings (page 17, lines 344 to 346).

Collectively, targeting *Itgb1l* expression inhibits scarring. Therefore, we are developing *Itgb1l* antisense oligodeoxynucleotides (ASOs). Our *in vitro* experiments have revealed three candidates for *Itgb1l* ASOs, and we will report on three *Itgb1l* ASO experiments.

Comment 2: *- Typo on page 18, line 363*

Response: We have corrected the error as indicated (page 18, line 360).

Comment 3: *- The manuscript is rather long. Please shorten the text.*

Response: We have shortened the text as requested.

Referee #3:

In this article the authors have explored the cell types and the molecular mechanisms that govern the inflammation driven scarring of the skin. Skin wound healing is a process involving inflammatory phase, proliferation phase and maturation phase. Lot happens during these phases starting with inflammatory cell infiltration, re-epithelialization and formation of granulation tissue and finally degradation of extracellular matrix. Wound healing is a complex process with actively involving cellular and molecular level changes in the wound microenvironment, but what happens during inflammation driven scarring, remains unclear. To address this authors have developed very interesting novel strategy, by combining high resolution spatial multiomics methods integrating spatial transcriptome utilizing 10 x Genomics Visium platform with temporal scRNA-seq datasets to identify new cell-cell communications and signaling during the wound repair process. Single cell RNA-Seq on murine skin wounds provided information about the temporal dynamic changes in the single cell populations and genes expressed in the wound. Also authors studied the gene expression heterogeneity in macrophages and fibroblasts involved in the wound healing. They have studied the cell-cell communications by cellchat at the skin wound sites. For Spatial transcriptional expression of inflammation related genes authors have utilized the Visium platform for 697 tissue spots at the murine wound site, which are data points equivalent to microdissection. The data generated from these two methods and PU.1^{-/-} mice that lack inflammatory response was combined and screened to identify the candidate genes involved in inflammation and scarring during skin wound. These experiments led to identification of nine genes 9 with potential involvement in inflammation-related scarring, including integrin beta-like 1 (Itgbl1) that is regulated by antagonistic interplay between TGF β 1 and IL1 β in the dermal fibroblasts . Further CRISPR-Cas technology to generate transgenic mice, authors have confirmed the role of Itgbl1 in skin wound healing and regulation of fibrogenesis and the Itgbl1-expressing cells were involved in collagen organization and are a pivotal part of the initiation of scarring at skin wound sites. Also, minority population of Acta2^{high}-expressing myofibroblasts appears to be involved in scarring in association with Itgbl1 expression. Overall, the new methodology and data presented by authors will help paving the way towards deeper understanding of skin wound healing and will help in development of new tools for clinical applications to study inflammation and fibrosis.

Major comments: The authors have integrated the spatial and temporal gene expression profiles and have shortlisted the potential genes involved in inflammation related scarring

during the wound healing. The authors conducted vigorous studies and have provided lot of information. There are major concerns that need to be addressed:

Comment 1: 1) Authors have discussed the gene expression changes in macrophages but have not detailed the crosstalk between the macrophage and the fibroblasts during early stages of wound healing. What are the macrophage-to-fibroblast interaction pathways that are involved in fibroblast activation? How macrophages are involved in the activation of fibroblasts. Did authors performed any experiments involving inhibiting TGF β 1 signaling and check its effect on fibroblast activation mediated by macrophages.

Response: The single-cell analysis results indicating the presence of cells (Fig 1) and the CellChat analysis results (Fig EV3A) revealed associations mainly with M1 (classically activated macrophages), M2, and M3 macrophage subclusters (alternatively activated polarized macrophages), as well as the F8 fibroblast cluster (which is highly likely to undergo subsequent phenotypic transformation into myofibroblasts). This crosstalk between macrophages and fibroblasts occurred during the early stages of wound healing (Day 3 post injury). Although the numerous interacting pathways suggested by the high-expressing genes of each macrophage (Fig EV1) and the CellChat results (Fig EV3B-3F) hinder clear predictions, we speculate that TGF β 1 (mainly associated with M1 and M3), CCL8 (mainly associated with M2), Spp1 (mainly associated with M3), and other factors influence activation of the F8 subcluster. The crosstalk between macrophages and fibroblasts during the early stages of wound healing is described in detail on page 12, lines 214 to 221.

Based on previous reports (Zhu H et al., *Arch Biochem Biophys*, 2022, PMID: 36183845), stimulation of neonatal cardiac fibroblasts with TGF β 1 induces the expression of ITGBL1, which is shown to be inhibited by the specific TGF β receptor I (ALK5) inhibitor SB431542. Therefore, we conducted confirmation experiments using murine-dermal skin-derived primary fibroblasts (MDF) and human-dermal skin-derived primary fibroblasts (HDF) using a A83-01 that is a potent inhibitor of TGF β type I receptor ALK5 kinase, type I nodal receptor ALK4 and type I nodal receptor ALK7 activation. As a result, it was confirmed that TGF β 1 induces the expression of *Itgbl1* (Appendix Fig S6B, C). However, despite expectations, the A 83-01 inhibitor did not suppress the expression of *Itgbl1*, although it did inhibit the expression of *Colla1* and *Col3a1* (positive controls) (see below Fig 2 for referee only). We carefully addressed the experimental points raised by the referee, but unexpected results were obtained. Although these results are intriguing, detailed investigation into TGF β 1 signal transduction is out of the scope of this paper. Therefore, we position this as

a newly derived research theme from our study's findings and would like to pursue it as the next research objective.

Figure 2 (referee only).

qPCR of *Itgbl1*, *Col1a1*, and *Col3a1* expression relative to that of *GAPDH* in TGFβ1-stimulated (100 pg/mL) in MDF (A) and HDF (B) treated with 1 μM of A 83-01 for 1 day ($n = 6$).

Data information: All values represent the mean. Unpaired *t*-test was used to generate the indicated *P* values; ** $P < 0.01$, *** $P < 0.001$, **** $P < 0.0001$

Comment 2: 2) Did author's study change in expression levels of type I collagen gene (*Col1a1*) using the Visium platform at day 7 and 14? After IL1 β treatment, was there any increase in the levels of collagen 3A1. Was there any effect on the ratio of collagen 1/collagen 3. Did author's confirmed IL1 β levels at later time points like day 14?

Response: We investigated the expression of *Col1a1* using the Visium platform on Days 3, 7, and 14 post injury. Our findings revealed that the expression of *Col1a1* at wound sites on Day 7 appeared to be increased compared with wound sites on Day 3 (see below Fig 3 for referee only). However, we encountered difficulty in precisely discerning differences in *Col1a1* expression levels between wound sites on Days 7 and 14 due to the high expression levels observed on both days. Therefore, we conducted further investigation of *Col1a1* expression levels using scRNA-Seq (Fig 2B).

Figure 3 (referee only). Spatial expression of *Col1a1* at skin wound sites on Days 3, 7, and 14 post injury according to Visium analysis.

W; wound sites; I; intact skin; M; muscle

We tested the expression of COL3A1 in HDF treated with IL1 β and found that its expression was significantly inhibited by IL1 β recombinant protein (Fig 6D and F).

We also investigated *Col1a1/Col3a1* gene expression at wound sites in WT and *Itgbl1*^{-/-} mice using qPCR and found that the *Col1a1/Col3a1* ratio did not change between the two groups (Appendix Fig S5). We speculate that the reduction in skin scarring among *Itgbl1*^{-/-} mice is related to aberrant modification of higher-order structures after collagen synthesis (e.g., fibrogenesis and collagen assembly). Thus, we hypothesize that *Itgbl1* plays a role in promoting fibrogenesis (Fig 7O).

We confirmed the expression of IL1 β using scRNA-Seq and ELISA methods (Appendix Fig S3D and F), and described these findings in the main text (page 19, lines 402 to 404).

Comment 3: *Minor comments:*

1) Please revise the manuscript to correct the typing errors; Page 4, Line 40, change reveled to revealed. Figure 7L- please correct assenbly to assembly. Page 18 line 363-its an incomplete sentence.

Response: We have corrected the indicated typos (page 4, line 38; page 18, line 360; and Figure 7O).

Comment 4: *2) Please specify in methods section, how many mice (biological and technical replicates) were utilized for scRNA-Seq and Visium platform. Are 40,000 cells from a wound sufficient to get to a conclusion, more sample size will help to address this question.*

Response: Regarding the details of single-cell analysis, we combined four wounds per sample for this analysis. This was because the number of single cells obtained from four wounds was most suitable for integrating the reagent conditions and three results.

As shown in Fig 1B, the cell count was highest at Day 7 and lowest at Day 3. According to histological analysis, cells accumulating at the wound site at Day 3 post injury (late inflammatory phase) are mainly neutrophils and macrophages. At Day 7 (proliferative phase), the number and types of cells increase, including fibroblasts involved in granulation tissue formation, endothelial cells, and epithelial cells involved in re-epithelialization. Considering these histological results, we believe our single-cell findings are sufficient to draw reliable conclusions.

Regarding Visium analysis, only one sample was used.

Details regarding these experimental methods and sample numbers have been added to the main text (page 31, line 655; and page 47, line 1002).

Dear Prof. Mori,

Thank you for submitting your revised manuscript. It has now been seen by two of the original referees.

As you can see, the referees find that the study is significantly improved during revision and recommend publication. However, I need you to address the points below before I can accept the manuscript.

- Please address the remaining minor concerns of referees #1 and #3.
- Please rename the 'Conflict of interests' section as 'Disclosure and Competing Interests Statement'
- Please remove 'Author Contributions' section from the manuscript text.
- Please make the dataset GSE234272 publicly available and remove the review access code from the manuscript text.
- We note the following regarding the Dataset EV legends: legends are only in Datasets EV1 and EV6. All datasets need to have legends and they should be included as a separate tab in each Excel file (all 10 datasets uploaded as a zip folder).
- Appendix file needs to be in PDF format. Page numbers are currently missing in the Table of Contents. Expanded View Datasets Legends should be removed from Appendix PDF and uploaded as separate tabs in each Dataset Excel file.
- All research articles submitted as revised versions must include a structured methods section that includes a Reagents and Tools Table followed by a Methods and Protocols section. Please see <https://www.embopress.org/page/journal/14693178/authorguide#structuredmethods> for further information.
- Source data files need to be saved in a scheme one figure/folder and then uploaded as .zip files. E.g. all the Source data files for figure 1 need to be saved in a single folder and this needs to be zipped and then uploaded as "SD figure 1.zip" file.
- During our routine image checks, we note a potential re-use between Figure 3B - and Figure 4B. Please clarify, also in the legends of both panels.
- Our production/data editors have asked you to clarify several points in the figure legends:
 - o Figure Legends (main + EV): 1. Please note that the exact p values are not provided in the legends of figures 5a, h, m; 6a-f; 7j, l, n.
 - o Please note that in figures 5a, h, m; there is a mismatch between the annotated p values in the figure legend and the annotated p values in the figure file that should be corrected.
 - o Please note that information related to n is missing in the legends of figures 1d; 2a-d; EV 2b.
 - o Although 'n' is provided, please describe the nature of entity for 'n' in the legends of figures 5a, h; 6a-f; 7l.
 - o Please note that for heatmap present in figures EV 3b; EV 4b; EV 5b; a numbered scale bar is not provided. This needs to be rectified.
 - o Please note that the scale bar needs to be defined for figure 7b.

Thank you again for giving us to consider your manuscript for EMBO Reports, I look forward to your minor revision.

Kind regards,

Deniz Senyilmaz Tiebe

--

Deniz Senyilmaz Tiebe, PhD
Senior Scientific Editor
EMBO Reports

Referee #1:

The authors have performed a large number of experiments to address the comments of the reviewers, and the revised manuscript is further improved. I just think that the wording regarding the importance of ITGBL1 for granulation tissue formation should be modified. It clearly plays a role in this process (although the precise function has to be determined in further studies), but it is not indispensable or essential as stated in the manuscript. I suggest to clarify this point and to write that it plays an important role. In fact, ITGBL1 obviously modifies the composition of the granulation tissue.

Referee #3:

Authors have adequately addressed the major comments that we raised. Regarding a minor comment, author provided sample size n=1 in the figure legend (Figure 3B) instead of the methods section. We suggest that the authors can incorporate it in the methods section.

Professor Bernd Pulverer
Editor-in-Chief
EMBO Reports

Re: Manuscript Number: EMBOR-2024-58791V2

30 September 2024

Dear Professor Pulverer,

Please find attached our revised manuscript entitled “Novel integrated multiomics analysis reveals a key role for integrin beta-like 1 in wound scarring”, by Sang-Eun Kim et al., which we request you to reconsider for publication as a Research Article in *EMBO Reports*.

We appreciate the positive feedback from the referees and the editor. Please find below our responses to all comments; changes to text in the revised manuscript are indicated using red font. We hope that the revised manuscript is now suitable for publication in *EMBO Reports*.

We look forward to hearing from you at your earliest convenience.

Yours sincerely,

Ryoichi Mori

Editor comments:

Dear Prof. Mori,

Thank you for submitting your revised manuscript. It has now been seen by two of the original referees.

As you can see, the referees find that the study is significantly improved during revision and recommend publication. However, I need you to address the points below before I can accept the manuscript.

Comment 1: *Please address the remaining minor concerns of referees #1 and #3.*

Response: We have included our responses to the reviewers' comments on page 6.

Comment 2: *Please rename the 'Conflict of interests' section as 'Disclosure and Competing Interests Statement'*

Response: We have renamed this section as indicated (page 37, line 793).

Comment 3: *Please remove 'Author Contributions' section from the manuscript text.*

Response: We have removed this section as indicated (page 37).

Comment 4: *Please make the dataset GSE234272 publicly available and remove the review access code from the manuscript text.*

Response: We have made our dataset publicly available (it can now be found at/requested from <https://www.ncbi.nlm.nih.gov/geo/query/acc.cgi?&acc=GSE234272>), as requested (page 36, lines 777 to 779).

Comment 5: *We note the following regarding the Dataset EV legends: legends are only in Datasets EV1 and EV6. All datasets need to have legends and they should be included as a separate tab in each Excel file (all 10 datasets uploaded as a zip folder).*

Response: We have added legends to ALL EV Datasets as requested.

Comment 6: *Appendix file needs to be in PDF format. Page numbers are currently missing in the Table of Contents. Expanded View Datasets Legends should be removed from Appendix PDF and uploaded as separate tabs in each Dataset Excel file.*

Response: We have modified the Appendix (made it PDF format) in and attended to all your other requests re this document.

Comment 7: *All research articles submitted as revised versions must include a structured methods section that includes a Reagents and Tools Table followed by a Methods and Protocols section. Please see <https://www.embopress.org/page/journal/14693178/authorguide#structuredmethods> for further information.*

Response: We have created the Reagents and Tools Table as indicated and uploaded this file (EMBOR-2024-58791V2_Reagents_Tools_Table).

Comment 8: *Source data files need to be saved in a scheme one figure/folder and then uploaded as .zip files. E.g. all the Source data files for figure 1 need to be saved in a single folder and this needs to be zipped and then uploaded as "SD figure 1.zip" file.*

Response: We have uploaded SD figures 1 to 4 and 6 as a zip file to the EMBO Rep website. However, we were unable to upload SD figures 5 and 7 due to their large file sizes. Could you please download them from the URL below? If you encounter any issues, please let us know.

URL:

<https://nudrive.nagasaki-u.ac.jp/public/865LwxMKUDkvr4FpnNGWlc1zOcCtn26IE2V92Ccn-0vU>

Password:

EMBOR-2024-58791V2

Comment 9: *During our routine image checks, we note a potential re-use between Figure 3B - and Figure 4B. Please clarify, also in the legends of both panels.*

Response: The H&E staining images in Figure 3B and Figure 4B to 4D were obtained using single sets of Visium data. We have mentioned this in the corresponding figure legends (page 44, lines 954 to 955; page 45, lines 966 to 967).

Comment 10: *Our production/data editors have asked you to clarify several points in the figure legends:*

Figure Legends (main + EV): 1. Please note that the exact p values are not provided in the legends of figures 5a, h, m; 6a-f; 7j, l, n.

Response: We have now added exact P values as indicated (page 45, lines 974 to 975; page 46, lines 990 to 991, 995 to 996, and 1008; page 47, lines 1018 to 1019, 1021, 1023 to 1025,

and 1027 to 1029; page 47, line 1031 to page 48, line 1033; page 48, lines 1036 to 1037; and page 49, lines 1062 to 1063, 1068 to 1069, and 1073 to 1074).

Comment 11: *Please note that in figures 5a, h, m; there is a mismatch between the annotated p values in the figure legend and the annotated p values in the figure file that should be corrected.*

Response: We have checked the figure legend and Figure 5 so that they now agree. We hope that our changes have addressed your comments.

Comment 12: *Please note that information related to n is missing in the legends of figures 1d; 2a-d; EV 2b.*

Response: We have now added the numbers of cells as requested (page 43, lines 929 to 933; page 44, lines 938 to 939, 943 to 945, and 947 to 948).

Comment 13: *Although 'n' is provided, please describe the nature of entity for 'n' in the legends of figures 5a, h; 6a-f; 7l.*

Response: We have described the nature of the entity for 'n' as requested (page 45, line 974; page 46, lines 990 and 1008; page 47, lines 1018, 1021, 1023, 1027, and 1031; page 48, lines 1035 to 1036).

Comment 14: *Please note that for heatmap present in figures EV 3b; EV 4b; EV 5b; a numbered scale bar is not provided. This needs to be rectified.*

Response: We have modified Figures EV3B, EV4B, and EV5B to add such a scale bar as requested.

Comment 15: *Please note that the scale bar needs to be defined for figure 7b.*

Response: We have now defined the scale bar as requested (page 48, line 1049).

Thank you again for giving us to consider your manuscript for EMBO Reports, I look forward to your minor revision.

Kind regards,

Deniz Senyilmaz Tiebe

Deniz Senyilmaz Tiebe, PhD

Senior Scientific Editor

EMBO Reports

Referee #1:

Comment 1: *The authors have performed a large number of experiments to address the comments of the reviewers, and the revised manuscript is further improved. I just think that the wording regarding the importance of ITGBL1 for granulation tissue formation should be modified. It clearly plays a role in this process (although the precise function has to be determined in further studies), but it is not indispensable or essential as stated in the manuscript. I suggest to clarify this point and to write that it plays an important role. In fact, ITGBL1 obviously modifies the composition of the granulation tissue.*

Response: Thank you for this insightful comment. We appreciate your suggestion regarding the wording of the role *Itgbll* plays in granulation tissue formation. We have revised the manuscript to clarify that although *Itgbll* plays an important role in this process and modifies the composition of the granulation tissue, it is not absolutely indispensable. We have made this modification as requested (page 6, line 68; page 19, line 398; page 21, line 446; page 22, line 459).

Referee #3:

Comment 1: *Authors have adequately addressed the major comments that we raised. Regarding a minor comment, author provided sample size $n=1$ in the figure legend (Figure 3B) instead of the methods section. We suggest that the authors can incorporate it in the methods section.*

Response: Thank you for this observation. We have incorporated the sample size information in the Materials and Methods section as requested (page 30, line 633).

Prof. Ryoichi Mori
Nagasaki University
Department of Tissue Repair and Regenerative Medical Science, Atomic Bomb Disease Institute
1-12-4 Sakamoto
Nagasaki 852-8523
Japan

Dear Prof. Mori,

Thank you for submitting your revised manuscript. I have now looked at everything and all is fine. Therefore, I am very pleased to accept your manuscript for publication in EMBO Reports.

Congratulations on a nice work!

Kind regards,

Deniz Senyilmaz Tiebe

--

Deniz Senyilmaz Tiebe, PhD
Senior Scientific Editor
EMBO Reports

--
